# Structures of *Tetrahymena thermophila* respiratory megacomplexes on the tubular mitochondrial cristae

Fangzhu Han[1,2,3], Yiqi Hu[1,2,3], Mengchen Wu[1,2], Zhaoxiang He[1,2], Hongtao Tian[1,2] & Long Zhou [1,2]✉

*Tetrahymena thermophila*, a classic ciliate model organism, has been shown to possess tubular mitochondrial cristae and highly divergent electron transport chain involving four transmembrane protein complexes (I–IV). Here we report cryo-EM structures of its ~8 MDa megacomplex $IV_2 + (I + III_2 + II)_2$, as well as a ~ 10.6 MDa megacomplex $(IV_2 + I + III_2 + II)_2$ at lower resolution. In megacomplex $IV_2 + (I + III_2 + II)_2$, each $CIV_2$ protomer associates one copy of supercomplex $I + III_2$ and one copy of CII, forming a half ring-shaped architecture that adapts to the membrane curvature of mitochondrial cristae. Megacomplex $(IV_2 + I + III_2 + II)_2$ defines the relative position between neighbouring half rings and maintains the proximity between $CIV_2$ and $CIII_2$ cytochrome *c* binding sites. Our findings expand the current understanding of divergence in eukaryotic electron transport chain organization and how it is related to mitochondrial morphology.

Oxidative phosphorylation is the energy-generating step of cellular respiration and takes place on the prokaryotic cytoplasmic membrane or the eukaryotic inner mitochondrial membrane (IMM). The energetically favorable oxidation of NADH or succinate by the electron transport chain (ETC) complexes I-IV (CI-CIV) first establishes a cross-membrane electrochemical gradient of protons, or the proton motive force (pmf), which is then utilized by the complex V (CV) to drive the synthesis of ATP[1]. Apart from working individually, ETC complexes can assemble into supercomplexes (SCs), including SC $I + III_2$[2,3], SC $III_2 + IV_{1/2}$[4–7], and the respirasome SC $I + III_2 + IV$[8–10], as well as megacomplexes (MCs) such as MC $I_2 + III_2 + IV_2$[11–13]. Among these, SC $I + III_2$ has been shown to be widely conserved across most eukaryotic clades, including the Opisthokonta (fungi and metazoan)[2], Archaeplastida (algae and plants)[13], Alveolata (ciliates and Apicomplexans)[3], and Euglenozoan (kinetoplastids and euglenids)[14], while further association of CIV differs in positions[15]. The term ETC megacomplexes describe the connection of two SC $I + III_2$ by oligomerization of CIII or CIV, which is in line with the formation of even higher-order assemblies such as respiratory strings or patches in a range of species[13,16–19].

However, structural description is limited to the mammalian MC $I_2 + III_2 + IV_2$, where two copies of CI are coupled by dimerization of CIII[11,12], while possible organization of the respiratory patch remains hypotheses. In addition to CI passing electrons from NADH to ubiquinone (Q), CII participates in the tricarboxylic acid (TCA) cycle by oxidizing succinate into fumarate and depositing electrons also to Q, thus serving as an alternative ETC entry point[20]. Although proposed as a megacomplex component[11] and/or respirasome assembly regulator[21–23], CII has never been structurally confirmed as part of any ETC supercomplex. Currently, reported eukaryotic CII structures are mostly limited to metazoans[24–26]. CII structural diversity in other eukaryotic clades has been investigated biochemically and genetically[27,28], but without reported structures.

Recent studies of SC $I + III_2$, $CIV_2$[3], and CV oligomers[29] of the model organism *Tetrahymena thermophila* (Tt) demonstrated their compositional and mechanistic divergences compared to the classic mammalian models, which likely resulted from structural adaptation to the tubular cristae of ciliate IMM[30]. Solubilization of the *T. thermophila* mitochondria membrane by mild detergents such as lauryl maltose

[1]Department of Biophysics, Zhejiang University School of Medicine, Hangzhou, Zhejiang Province 310058, China. [2]Department of Critical Care Medicine of Sir Run Run Shaw Hospital, Zhejiang University School of Medicine, Hangzhou, Zhejiang Province 310058, China. [3]These authors contributed equally: Fangzhu Han, Yiqi Hu. ✉e-mail: longzhou@zju.edu.cn

neopentyl glycol (LMNG) and digitonin results in three separable bands with CI activity on blue native PAGE (BN-PAGE) (Supplementary Fig. 1). The lowest band corresponds to individual Tt-CIV$_2$, Tt-SC I + III$_2$ and Tt-CV$_2$ complexes[3,29], while the two upper bands likely correspond to the higher-order organization of different stoichiometries.

In this study, by solving the structures of *T. thermophila* megacomplex IV$_2$ + (I + III$_2$ + II) (Tt-MC IV$_2$ + (I + III$_2$ + II)$_2$), we reveal that the obligatory Tt-CIV$_2$ serves as the central coupler linking two Tt-SC I + III$_2$ and two Tt-CII into a half ring-shaped megacomplex adapted to the cristae IMM curvature. Our Tt-MC IV$_2$ + (I + III$_2$ + II)$_2$ structure agrees with a recent study of the ~5.8 MDa Tt-SC IV$_2$ + I + III$_2$ + II on multiple aspects, including overall organization, membrane curvature adaptation, Tt-CII structure, and inter-complex interaction sites[31]. We also demonstrate, via a low-resolution reconstruction of Tt-MC (IV$_2$ + I + III$_2$ + II)$_2$, how two megacomplex half rings associate into a respiratory patch to maintain proximity between cytochrome *c* (cyt *c*) binding sites of Tt-CIII and Tt-CIV.

## Results

### Overall structures of *T. thermophila* ETC megacomplexes

The two BN-PAGE upper bands of LMNG-solubilized *T. thermophila* mitochondria membrane was purified into a single sample using sucrose gradient ultracentrifugation and size-exclusion chromatography (Supplementary Fig. 1a–d). Spectroscopic assays monitoring the oxidation of NADH as well as the reduction and oxidation of Tt-cyt *c* confirmed that the sample was capable of electron transport through both NADH:O$_2$ and succinate:O$_2$ pathways at expected rates (Fig. 1d–h and Supplementary Figs. 2a–c, 3c, d)[3,32]. The lower NADH oxidation rate (Fig. 1d) here compared to that reported using the *T. thermophila* mitochondria membrane may be caused by possible inhibitory effects of the solubilization detergents[3]. More likely, this is due to the fact that the initial fast phase of NADH oxidation (0–50 s) is difficult to capture accurately with a high enzyme concentration of 10 nM (Supplementary Fig. 3c). When this is lowered to 2 nM, the initial fast phase of NADH oxidation expands to around 300 s, and the NADH oxidation rate reaches above 400 nmole/min/mg protein similar to the literature value (Supplementary Fig. 3a, b)[3]. The succinate oxidation rate measured from the cyt *c* reduction curve is ~47 nmole/min/mg protein, roughly consistent with the ~30 nmole/min/mg protein succinate-dependent oxygen consumption rate measured with *T. thermophila* cells[33]. NADH and succinate-dependent activities could be inhibited by CI inhibitor rotenone and CII inhibitor malonate, respectively. Moreover, both activities could also be inhibited by CIII$_2$ inhibitor antimycin A and CIV inhibitor sodium azide, demonstrating that electron transport initiated by Tt-CI and Tt-CII were tightly coordinated with Tt-CIII$_2$ and Tt-CIV activities in our purified assemblies of Tt-SC IV$_2$ + I + III$_2$ + II and Tt-MC IV$_2$ + (I + III$_2$ + II)$_2$ (Fig. 1d–h and Supplementary Fig. 3c, d).

Single-particle cryo-electron microscopy (Cryo-EM) revealed the identities of the two upper bands as Tt-SC IV$_2$ + I + III$_2$ + II and Tt-MC IV$_2$ + (I + III$_2$ + II)$_2$, respectively (Supplementary Fig. 4). Corresponding to 77 and 23% of the total particles, these two assemblies were then refined to overall resolutions of 2.89 and 2.96 Å (Supplementary Figs. 4, 5a and Supplementary Table 1). To better resolve the two-fold symmetric megacomplex, focused refinements were performed on different Tt-SC I + III$_2$ and Tt-CIV parts of the re-extracted asymmetric unit IV + I + III$_2$ + II, giving resolutions from 2.8 to 2.96 Å (Supplementary Figs. 4 and 5a–f and Supplementary Table 1). To better resolve Tt-CII, which is prone to dissociation in LMNG, its focused refinement was performed with the combination of Tt-SC IV$_2$ + I + III$_2$ + II and Tt-MC IV$_2$ + (I + III$_2$ + II)$_2$ particles (Supplementary Fig. 4). A well-resolved 3.26 Å map was produced (Supplementary Figs. 4, 5g and Supplementary Table 1), indicating that Tt-CII adopts similar conformations in the two assemblies. Focused refinement maps were then combined into a composite map of dimeric Tt-MC IV$_2$ + (I + III$_2$ + II)$_2$ (Supplementary Figs. 4, 5a and Supplementary Table 1).

Tt-MC IV$_2$ + (I + III$_2$ + II)$_2$ is a ~8 MDa megacomplex encompassing the entire ETC of *T. thermophila*. A total of 364 subunits are identified, mostly already assigned by blasting the density-derived query sequence against the *T. thermophila* proteome from UniProt in the previous study (Fig. 1a, b, Supplementary Data 1, and Supplementary Movie 1)[3]. Tt-CIV$_2$ serves as a central coupler in this megacomplex, with each protomer associating one copy of Tt-SC I + III$_2$ and one copy of Tt-CII. Transmembrane (TM) helices (TMHs) from Tt-CI to Tt-CIV form a continuous patch of membrane domain flanked by two copies of Tt-CIII$_2$, exhibiting a ~160° curvature when the tubular cristae is viewed axially (Fig. 1a). This is in accordance with the ~50° curvatures of individual Tt-CIV$_2$ and Tt-SC I + III$_2$[3], as well as the known ~40 nm diameter of ciliate cristae cross-section[30]. Formation of such half ring-shaped megacomplex clearly set orientations for each of its components on the IMM. When viewed perpendicular to the IMM, the long axis of the ellipse-shaped Tt-CIV$_2$ is slanted ~35° with respect to the axial direction of the cristae tube (Fig. 2a).

We were also able to isolate a minor particle set corresponding to Tt-MC (IV$_2$ + I + III$_2$ + II)$_2$ in a separate cryo-EM dataset using digitonin-solubilization, focused refinement of which resulted in 4.18 and 6.77 Å for its two Tt-SC IV$_2$ + I + III$_2$ + II parts (Supplementary Figs. 4, 5). Although the resolution prevents building an atomic model, it is clear that the two supercomplexes belong to neighboring megacomplex half rings (Supplementary Fig. 8c, d). At the megacomplex-megacomplex interface, Tt-SC I + III$_2$ contact Tt-CIV$_2$ from the adjacent megacomplex, while the two Tt-CIIs face each other directly (Supplementary Fig. 8a, b and Supplementary Movie 2). This provides a glimpse into the structure of ciliate respiratory patch[16] and suggests that it contains at least stacks of ETC half rings that sheath the tubular cristae (Fig. 1a). Although the compositional organization of possible full ring involving Tt-CV is currently unknown, comparable axial dimensions are observed for Tt-MC IV$_2$ + (I + III$_2$ + II)$_2$ and Tt-CV$_4$ (Supplementary Fig. 9a, b), raising the hypothesis that one-half ring of Tt-MC IV$_2$ + (I + III$_2$ + II)$_2$ corresponds to two axially packed Tt-CV$_2$ in their strip of dimer row along the cristae tube[29,30].

### Organization of Tt-MC IV$_2$ + (I + III$_2$ + II)$_2$

The obligatory Tt-SC I + III$_2$ largely shares the conserved architecture of SC I + III$_2$ across different species[15], apart from its unique membrane domain curvature[3]. The tunnel for Q/QH$_2$ to enter and exit CI opens to the concave side of CI MA bordered by CIII$_2$ in SC I + III$_2$ (Fig. 1c)[15]. However, while the mammalian respirasome monomeric CIV associates SC I + III$_2$ via its CI MA toe (in analogy to human feet)[8], in Tt-MC IV$_2$ + (I + III$_2$ + II)$_2$ the constitutively dimeric Tt-CIV$_2$ associates the Tt-CI MA heel from the side opposing its Q tunnel (Figs. 1c, 2a, b)[34]. Moreover, Tt-CII participates in Tt-megacomplex formation in contrast to the mammalian case[11], wedging into the gap between Tt-CIV$_2$ and Tt-CI MA toe. In this way, Tt-CIV$_2$ contacts Tt-CI and Tt-CII via its two sides where the mitochondrial carrier subunits COXMC3 and COXMC1-2 sit, respectively (Fig. 2a, b and Supplementary Data 1)[3]. Dimerization of Tt-CIV, therefore, couples two copies of Tt-SC I + III$_2$ and Tt-CII each (Fig. 1a), thus playing a central role in megacomplex formation.

Such drastic re-arrangement of megacomplex architecture compared to that of the mammals (Fig. 1c) is possible due to not only the highly divergent Tt-CIV$_2$ but also the existence of Tt-specific CI accessory subunits and extensions (Supplementary Fig. 10a, b). Tt-CIV$_2$ is a ~2.7 MDa complex[3], much larger than bacterial[35], archaeplastidan[5], and opisthokont[36–38] CIV, ranging from 135 to 213 kDa. It possesses a total of 43 subunits not previously identified as CIV components[3]. CIV subunits COX7A, COX7C, and COX8B (Supplementary Data 1), which interact with CI MA toe and CIII$_2$ and in mammalian respirasome[8], are either not present in Tt-CIV$_2$ (COX8B) or buried deeply inside by other Tt-specific subunits (COX7A and COX7)[3]. In addition, the Tt-specific CI toe bridge structure embraces Tt-CIII$_2$ in Tt-SC I + III$_2$ and precludes canonical respirasome assembly via the interaction between CI MA toe

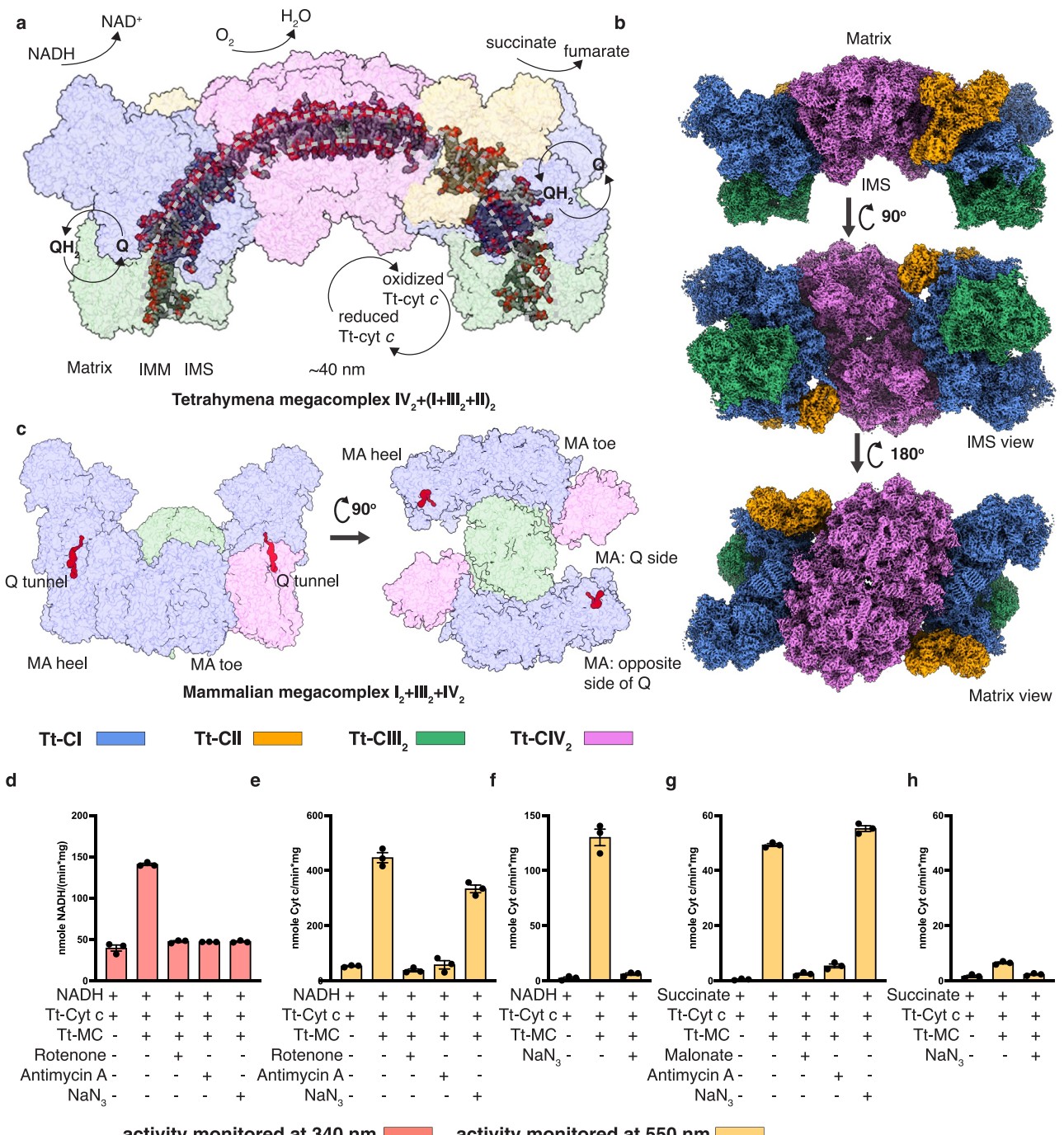

**Fig. 1 | Overall structure of *T. thermophila*'s electron transport chain megacomplex IV$_2$ + (I + III$_2$ + II)$_2$. a** Axial view of Tt-MC IV$_2$ + (I + III$_2$ + II)$_2$ overlaid with the cross-section of the ciliate tubular cristae. Individual ETC complexes are shown on transparent surfaces and colored as indicated. Phospholipids are shown as spheres and indicate the ~160° curvature of the megacomplex membrane domain. The arrows mark approximate ubiquinone and cyt *c* redox sites. **b** Cryo-EM map of Tt-MC IV$_2$ + (I + III$_2$ + II)$_2$ colored by individual complex as in (**a**) and viewed from different directions. **c** Atomic model of mammalian MC I$_2$ + III$_2$ + IV$_2$ (PDB: 5XTI)[11], shown as in (**a**). Q tunnels for mammalian CI are shown as red surfaces. CI MA heel and toe, as well as the two MA sides differentiated by the Q tunnel, are labeled. Certain subunits of CI PA are removed for clarity in the right panel. **d–h** NADH:O$_2$ (**d–f**) and succinate:O$_2$ (**g–h**) oxidoreductase activities of Tt-MC IV$_2$ + (I + III$_2$ + II)$_2$,

in the absence or presence of specific inhibitors of CI-CIV, measured using 10 nM Tt-megacomplex sample. Activities are monitored both at 340 nm by NADH oxidation (**d**) and at 550 nm by cyt *c* reduction (**e**, **g**) and oxidation (**f**, **h**). Cyt *c* reduction and oxidation rates are calculated from different phases of the same kinetic curves: **e**, **f** here are based on the kinetic curve in Supplementary Fig. 3c right panel, **g**, **h** here are based on the kinetic curve in Supplementary Fig. 3d. Cyt *c* reduction rates in (**e**, **g**) here are based on the initial linear increasing phases between 50 and 150 s, while cyt *c* oxidation rates in (**f**, **h**) are based on the last linear decline phases (see Methods for details). Values are averages of three technical measurements from a single purified sample ± standard error of mean (SEM). Source data are provided as a Source Data file.

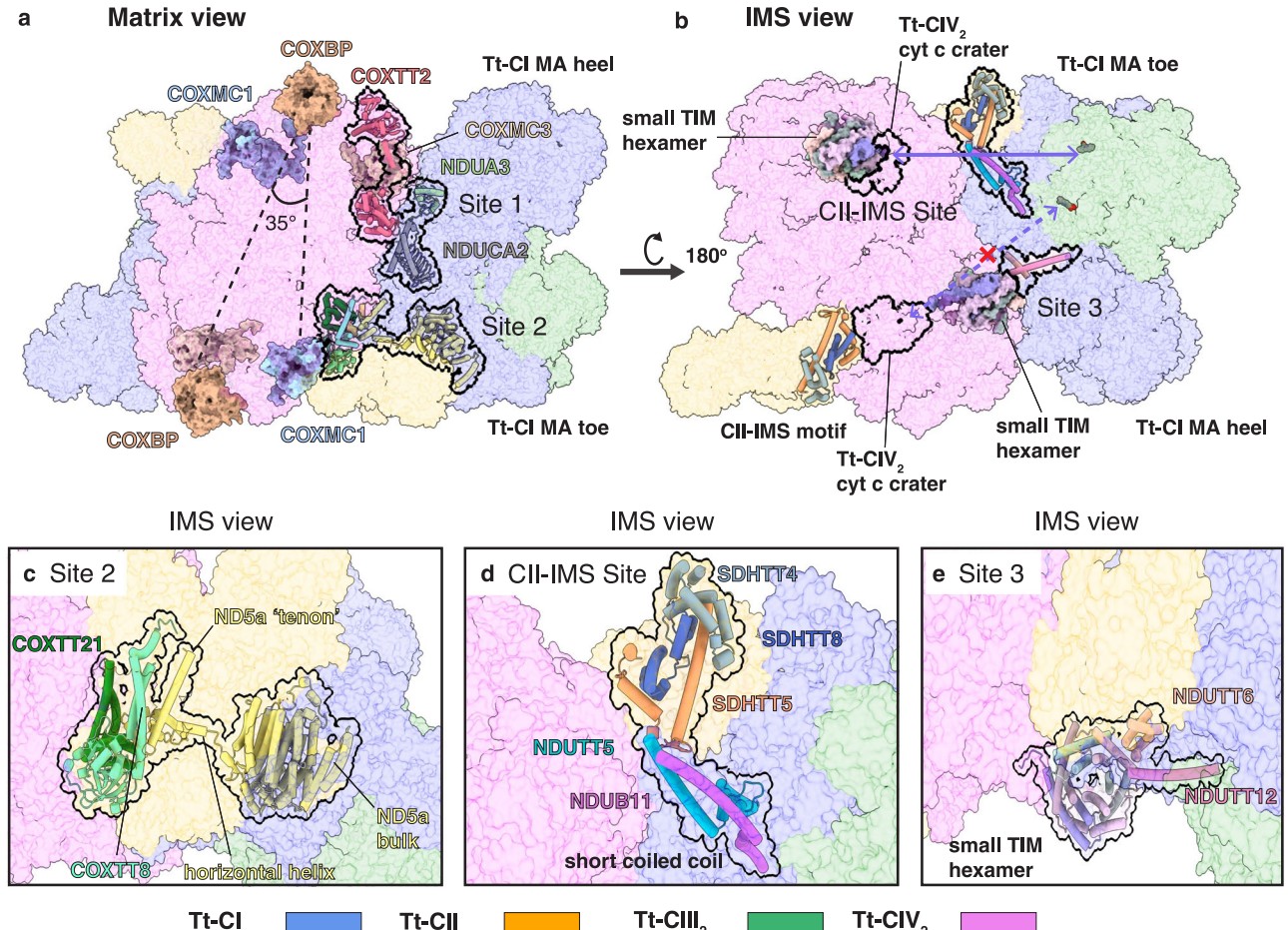

**Fig. 2 | Interaction sites within Tt-MC IV$_2$ + (I + III$_2$ + II)$_2$. a, b** Tt-CI-CIV$_2$ interaction sites 1–3 and Tt-CI-CII IMS interaction site with individual complexes shown as colored transparent surfaces, viewed from the matrix (**a**) and the IMS (**b**) side. Key subunits are shown as cylindrical cartoons and highlighted by black outlines. Subunits used to identify positions within individual complexes are shown as solid surfaces. The toe and heel ends of Tt-CI are labeled. In (**b**), the two cyt *c* craters of

Tt-CIV$_2$ are highlighted by black outlines. The kinetically most favorable distance for cyt *c* diffusion within a single Tt-MC IV$_2$ + (I + III$_2$ + II)$_2$ is marked by a solid blue double arrow, while the unfavorable one due to structural obstruction of the small TIM hexamer domain is identified by a dashed blue double arrow and a cross mark. **c**–**e** Zoom in of Tt-CI-CIV$_2$ interaction site 2 (**c**), site 3 (**e**), and the Tt-CI-CII site (**d**) viewed from the IMS side.

and Tt-CIV (Fig. 3a and Supplementary Fig. 10a, b)[8]. The significantly augmented Tt-CIV$_2$ thereby has to bind Tt-CI via a larger interface on its MA side opposing Q tunnel, which is at the outer rim of mammalian respirasome and megacomplex without contacting any other ETC complex (Fig. 2c)[8,11].

Although the relative position between Tt-CI and Tt-CIV$_2$ is divergent in Tt-MC IV$_2$ + (I + III$_2$ + II)$_2$, subunit NDUTT4 at Tt-CI's MA toe does contact Tt-CIV$_2$ from the neighboring megacomplex via subunit COXMC2, as revealed by Tt-MC (IV$_2$ + I + III$_2$ + II)$_2$ structure (Supplementary Fig. 8f and Supplementary Data 1). Therefore, contact between CIV and CI MA toe is preserved not within the half ring structure of Tt-MC IV$_2$ + (I + III$_2$ + II)$_2$, but across stacks of rings with defined register on the IMM cristae. Conservation of such interaction across eukaryotic clades, albeit structural rearrangements, emphasizes the physiological importance of spatial proximity between cyt *c* redox sites of CIII$_2$ and CIV. Decreasing such distance via supercomplex formation could decrease cyt *c*'s diffusion volume needed to achieve electron transfer and therefore provide a kinetic advantage over individual complexes, as demonstrated by theoretical[39] and mutational[40] approaches. Restricting cyt *c*'s 3D diffusion with the bulk matrix phase to electrostatically guided 2D diffusion on a supercomplex surface can further increase its turnover, as shown structurally in yeast SC III$_2$ + IV[6]. In Tt-MC (IV$_2$ + I + III$_2$ + II)$_2$, a direct distance of 97 Å is measured between the two cyt *c* redox sites across neighboring

megacomplexes, while within a single Tt-MC IV$_2$ + (I + III$_2$ + II)$_2$ the distance increases to 126 and 170 Å (Supplementary Fig. 8e). Although the shortest distance of 97 Å is slightly increased compared to 85, 61, and 70 Å in mammalian[4], yeast[7], and plant[5] SC III$_2$ + IV$_{1/2}$ structures, it can still enhance the cyt *c* turnover, considering that the tubular cristae of ciliates provide a smaller restrictive diffusion volume than the common lamellar cristae of opisthokonts[39].

Three main interaction sites can be found between Tt-CIV$_2$ and Tt-CI (Fig. 2). On the matrix side of IMM, ~200 residues on Tt-CIV$_2$ COXTT2 N-terminal (NT), previously invisible in individual Tt-CIV$_2$ structure[3], is now resolved as an all α-helical globular domain, fitting into a concave surface formed by the Tt-specific NT loop extension of NDUA3 and the C-terminal (CT) helix of NDUCA2 belonging to the γ carbonic anhydrase (γCA) domain (Fig. 2a and Supplementary Data 1). In the IMM, ~160 residues on the NT of Tt-CI core subunit ND5a, previously un-resolved in individual Tt-SC I + III$_2$, is now resolved in Tt-MC (IV$_2$ + I + III$_2$ + II)$_2$ from the digitonin dataset and fits into the concave surface formed by COXTT21 and COXTT8 (Supplementary Data 1). A short, horizontal helix in IMM traverses across the gap between Tt-CIV$_2$ and Tt-CI MA toe to connect the ND5a NT domain to its bulk portion from behind Tt-CII, which possibly helps to define the openness of the gap and how deep Tt-CII can wedge in (Fig. 2c). Acting like the mortise and tenon junctions of woodworks, these two domains of subunits COXTT2 and ND5a stabilize the Tt-CI to Tt-CIV$_2$ connection and

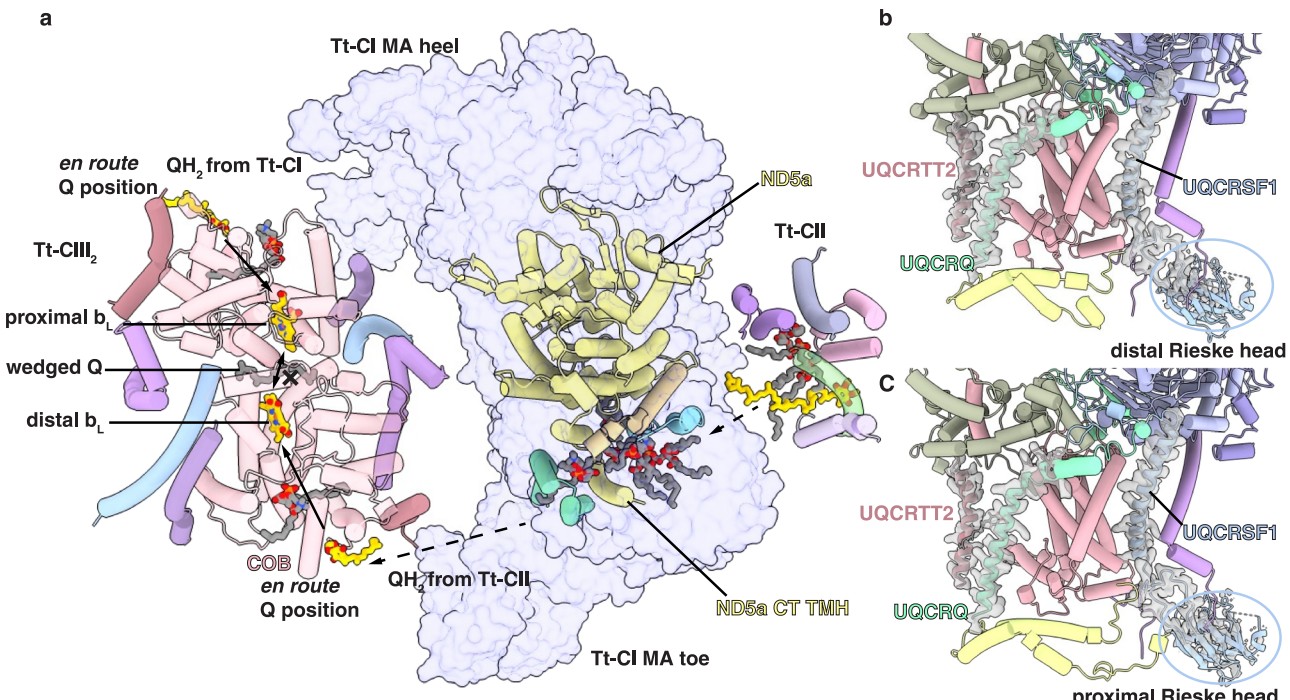

**Fig. 3 | Symmetry re-establishment of Tt-CIII$_2$ within Tt-MC IV$_2$ + (I + III$_2$ + II)$_2$.**
**a** Representation of the QH$_2$ diffusion direction from Tt-CII to the distal Q cavity of Tt-CIII$_2$. Key subunits are shown as cylindrical cartoon and colored by subunits. The membrane domain of Tt-CI is shown as a transparent surface, with its toe and heel ends labeled. Heme b$_L$, ubiquinone in Tt-CII, Tt-CIII$_2$ en route Q positions and Tt-CIII$_2$ wedged Q position and lipids along the hydrophobic pathway are shown as atomic sticks. Definitions of the distal and proximal sides of Tt-CIII$_2$ are labeled by the two b$_L$ hemes. **b, c** Densities for subunit UQCRTT2 TMH, UQCRQ TMH, and the UQCRFS1 Rieske head domain are shown for both Tt-CIII$_2$ distal (**b**) and proximal (**c**) protomers as gray transparent surfaces. Note that these structural elements display distinct density qualities for the two Tt-CIII$_2$ protomers in Tt-SC I + III$_2$.

become structurally ordered themselves. The third interaction site on the IMS involves COXTIM5 and COXTIM2 of the Tt-CIV$_2$ small TIM hexamer domain bridging over to NDUTT12 and NDUTT6 of Tt-CI (Fig. 2b, e and Supplementary Data 1). This IMS bridge is not conserved in Opisthokonta and poses a significant structural barrier between the nearby Tt-CIV$_2$ cyt $c$ crater and the Tt-CIII$_2$ cyt $c$ reduction site 126 Å apart. Its existence could increase the local structural crowdedness, thereby negatively interfere with the diffusion efficiency of cyt $c$ between the two sites (Supplementary Fig. 8e)[41].

## Tt-CII is a structurally divergent Type-D succinate dehydrogenase

In contrast to mammalian CII, which does not participate in supercomplex or megacomplex formation[11], Tt-CII is clearly incorporated into *T. thermophila* megacomplex as revealed by the Tt-MC IV$_2$ + (I + III$_2$ + II)$_2$ structure. The Y-shaped Tt-CII is a 288 KDa TM complex with eight hydrophilic and seven TM subunits, therefore structurally more complicated than the canonical four-subunit, ~130 KDa succinate dehydrogenases (SDHs) of *E.coli*[42] and opisthokonts (Fig. 4a, b and Supplementary Fig. 11a, b)[24,25]. SDHs and the related quinol:fumarate reductases (QFRs) have been previously classified into type A-F based on the numbers of TM subunit(s) and heme group(s)[43–46]. Tt-CII contains two core TM subunits, SDHC and SDHD (Fig. 4a), but no heme, therefore most likely belongs to the small subfamily of type-D SDH. Position of the in-membrane heme b found in the mammalian type-C CII is taken by cardiolipin in Tt-CII (Fig. 4d). Lack of heme b in a wild-type SDH confirms the idea that it is absolutely required neither as an SDH/QFR structural stabilizer[47] nor as a temporary Q reduction electron sink[42,48] for at least the ciliates. The classic six-TMH bundle formed by SDHC and SDHD in type-C[24,25,42] and -D[49] SDHs is only partially conserved in Tt-CII, mostly around the Q$_P$ site occupied by clear Q head density (Fig. 4d and Supplementary Fig. 12c).

Subunits SDHA and SDHB form one of the two hydrophilic heads of Tt-CII, highly conserved to known CII structures with the FAD and three iron-sulfur clusters in place (Fig. 3b, c and Supplementary Fig. 12b, d–f). A second Tt-specific hydrophilic domain is formed mainly by the Tt-specific SDHTT1 subunit, which composes an NT Rossmann fold and a CT α-helical domain coordinating a heme group at the Tt-CII and CIV$_2$ interface (Fig. 4a, e, Supplementary Fig. 12a, and Supplementary Data 1). The presence of a single thioether linkage to SDHTT1_Cys[321] and the lack of a hydroxyethylfarnesyl group identify it as a c-type heme. The recent study of Tt-SC IV$_2$ + I + III$_2$ + II deconvoluted its merged b- and c-type absorption peaks and identified an absorption maximum at 556 nm apart from heme b$_L$, b$_H$, and c$_1$, in line with the presence of an additional c-type heme[50]. Therefore, we tentatively assign this non-canonical heme as a heme c. It locates far away from the iron-sulfur clusters beyond direct electron transfer distance (Fig. 4e), therefore not likely involved in electron transport from succinate to Q by Tt-CII. Single TMH subunits SDHTT6 and SDHTT9 form a separate membrane domain beneath the second hydrophilic head of SDHTT1 and structurally anchor it onto the IMM (Fig. 4a and Supplementary Data 1). The two hydrophilic heads are connected by Tt-SDHTT3, a multi-helix bundle subunit parallelly lining the matrix side of IMM (Fig. 4a and Supplementary Data 1).

Tt-CII associates Tt-SC IV + I + III$_2$ at multiple sites. Most prominently, an IMS domain not present in any known CII structures, is formed by SDHTT4, 5, 8 and hydrophilic parts of SDHTT2, 7 and 10 (Fig. 2b and Supplementary Data 1). COXTT8, a key subunit involved in connecting Tt-CIV$_2$ to the NT domain of Tt-CI ND5a, now polarly contacts CT helical hairpins of SDHTT5 and SDHTT8 of this IMS domain (Supplementary Data 1). Notably, the NT helix of SDHTT5 interacts with the short coiled-coil of NDUB11 and NDUTT5 protruding into the IMS, a conspicuous structural feature of Tt-CI with unknown function in the previous study (Fig. 2b, d)[3].

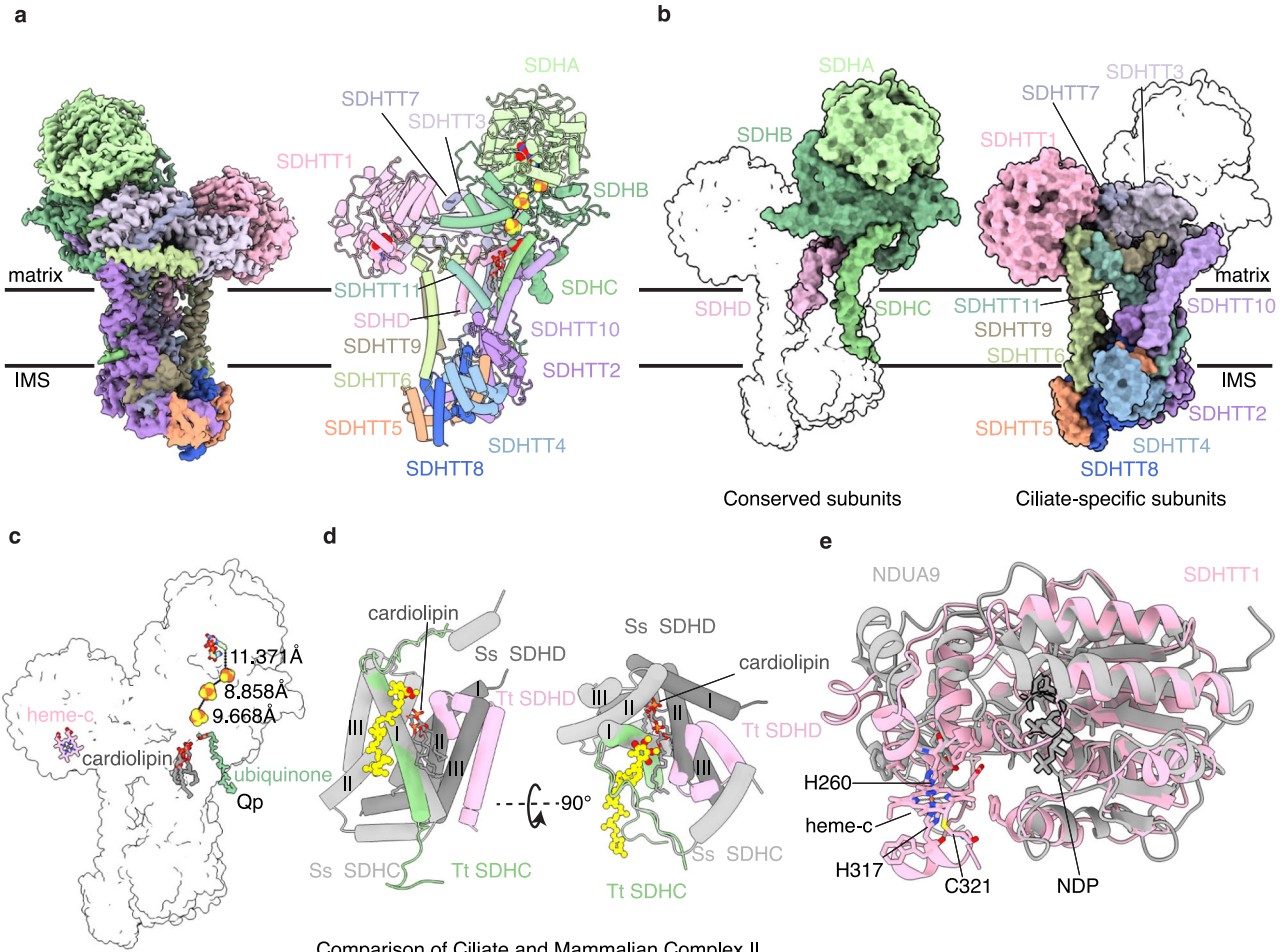

**Fig. 4 | Structure of Tt-CII. a** Cryo-EM map (left) and atomic model (right) of Tt-CII colored by individual subunits. **b** Structural silhouette of Tt-CII, with subunits conserved as in Opisthokonta (left) or ciliate-specific (right) shown in solid surfaces. **c** Prosthetic groups of Tt-CII. The FeS clusters are shown as atomic spheres and colored by atoms. The FAD, Qp site ubiquinone, an interfacial heme c between Tt-CII and Tt-CIV$_2$, and a cardiolipin molecule in place of the heme b in type A−C SDHs are shown as atomic sticks. Distances between the electron-transporting FAD and FeS clusters are labeled. **d** Structural superposition of *T. thermophila* (colored by subunits) and *Sus scrofa* (colored by different shades of gray, PDB: 1ZOY)[24] CIIs' TM subunits SDHC and SDHD. TMH 1−6 of the classic six-helix bundle of mammalian CII membrane domain are labeled by Roman numbers. The Qp site ubiquinone is shown as an atomic stick with its map density, the cardiolipin in place of the heme b in type A−C SDHs is shown as an atomic stick. **e** Structural conservation of Tt-CII subunit SDHTT1 (pink) and Tt-CI subunit NDUA9 (gray). NDP group of NDUA9 and the interfacial heme c SDHTT1 with its coordinating residues are shown as atomic sticks.

## Ciliate-specific subunits of Tt-CIV$_2$ contribute to megacomplex assembly

The highly divergent Tt-CIV$_2$ contains 43 ciliate-specific subunits, as summarized previously, surrounding three core subunits and six conserved subunits within the IMM and from both IMS and matrix sides (Supplementary Data 1)[3]. A portion of these subunits form the largely augmented dimer interface of Tt-CIV$_2$, including the single TMH subunit COXTT13 topologically superposing with NDUA4, the mammalian respirasome assembly factor (Supplementary Fig. 13a and Supplementary Data 1)[31,38]. The ciliate-specific NT helix-loop-helix extensions of both NDUA4 copies are spatially adjacent to each other on the dimer interface and together line the lipid-filled central cavity of Tt-CIV$_2$ from the matrix side parallelly to the IMM (Supplementary Fig. 13a). Likewise, TMHs of subunits COXTT6, COXTT8, COXTT9, COXTT20, COXTT21, COXTT23, and the Fe$_2$S$_2$ cluster containing subunit COXFS are like pillars standing in the hall of central cavity shared by the two Tt-CIV protomers, while hydrophilic domains of subunits COXTT6 and COXFS form additional dimerization interfaces in the IMS together with COXTT5, COXTT19 and the acyl carrier protein COXAC (Supplementary Fig. 13d).

A majority of the rest ciliate-specific subunits are found on the Tt-CIV$_2$'s interfaces to Tt-CI MA heel and Tt-CII within Tt-MC

IV$_2$ + (I + III$_2$ + II)$_2$, as well as to Tt-CI MA toe of the neighboring megacomplex. For example, three mitochondrial carriers COXMC1-3 are identified on the peripheral of Tt-CIV$_2$, of which COXMC1 and 2 form a heterodimer and are both annotated as 2-oxoglutarate carriers (SLC25A11) in UniProt database[3]. COXMC1 and 2 are kept in different states, with their central substrate cavities opening to the matrix and IMS sides, respectively. In this way, COXMC1 TMH2 comes in close contact with Tt-CII SDHTT1 on the matrix side and provides an additional Tt-CII attachment site to the megacomplex (Supplementary Fig. 13e). Meanwhile, the three outward-facing COXMC2 TMHs are kept in positions where they can easily contact Tt-CI's NDUB2 and NDUTT4 of the neighboring Tt-MC IV$_2$ + (I + III$_2$ + II), thereby creating an additional interaction site across the two megacomplexes in Tt-MC (IV$_2$ + I + III$_2$ + II)$_2$ apart from dimerization of Tt-CII's IMS domain (Supplementary Fig. 13e).

A third group of ciliate-specific subunits, including COX6BL, COX17L, COXTT25, and extensions of core subunits, are involved in the formation of Tt-CIV$_2$'s charge-swapped cyt c crater[3]. Notably, a TIM8$_3$-TIM13$_3$ chaperone-like domain formed by subunits COXTIM1-6 next to the cyt c crater sterically blocks the two Tt-SC I + III$_2$ copies from moving closer to the cristae center by its close proximity to Tt-CI's NDUTT12, therefore maintaining the structural integrity of the

megacomplex as well as the membrane curvature (Supplementary Fig. 13f and Supplementary Data 1). The above observations are in line with several structural studies of ETC from different organisms that individual enzyme homologs such as the γ-carbonic anhydrase[51–53], acyl carrier[3,54], and formate dehydrogenase[55] can associate and become parts of ETC complexes for energetic purposes or purely using them as structural scaffolds. Most of these homologs lost their original activities during evolution and instead served only stabilizing or assembly roles in ETC complex or megacomplex formation.

**Symmetry re-establishment of Tt-CIII$_2$ in Tt-MC IV$_2$+(I+III$_2$+II)$_2$**
The dimeric CIII$_2$ employs a Q-cycle mechanism to accomplish bifurcated electron transfer, where a ~ 60° rotation of the UQCRFS1 Rieske head Fe$_2$S$_2$ cluster is required to ferry electron from QH$_2$ at the Q$_P$ site towards heme c$_1$[56–59]. However, in the obligatory Tt-SC I+III$_2$, the Rieske head near Tt-CI MA toe (defined as distal) is sterically locked by the extensive CI-CIII$_2$ interactions and incapable of such swinging motion, as demonstrated by its much clearer electron density compared to the Rieske head near Tt-CI Q tunnel (defined as proximal)[3]. Moreover, an edge-to-edge distance between the two b$_L$ hemes, measured between C2B carbon atoms (using PDB atomic nomenclature of ligand HEM) of the two porphyrin rings[60], is 17.5 Å in the current Tt-MC IV$_2$+(I+III$_2$+II)$_2$ structure, contrasted by that of 13.3 Å in the recent high-resolution structure of *Arabidopsis thaliana* SC I+III$_2$ (PDB 8BPX)[61]. The generally accepted distance limit for direct electron transfer is 14 Å[62], therefore, electrons cannot transfer across the two Tt-CIII$_2$ protomers in Tt-MC IV$_2$+(I+III$_2$+II)$_2$. The increased inter-b$_L$ heme distance and symmetry breaking regarding Rieske head mobility suggest that in Tt-SC I+III$_2$, functional electron pathway exists only in the proximal Tt-CIII$_2$ protomer and transports electron from the Q$_P$ site in the proximal Q cavity to the Q$_N$ site in the distal Q cavity (Supplementary Fig. 14a, b)[3].

In Tt-SC IV$_2$+I+III$_2$+II and Tt-MC IV$_2$+(I+III$_2$+II)$_2$ however, both Rieske heads lack clear density, thereby likely capable of the swinging motion required for electron transfer towards cyt c (Fig. 3b)[59]. Moreover, a previously unidentified TM subunit existing only in the distal position is now coined subunit UQCRTT2 (Supplementary Data 1), and is present with equally clear densities in both positions (Fig. 3b and Supplementary Fig. 12m). Similarly, clear densities are also observed for the TMHs of both UQCRQ subunits in Tt-MC IV$_2$+(I+III$_2$+II)$_2$ (Supplementary Fig. 12n), while in Tt-SC I+III$_2$ only the proximal UQCRQ TMH is visible[3]. Taken together, structural symmetry between two Tt-CIII$_2$ protomers that was broken in Tt-SC I+III$_2$ is now re-established in Tt-MC IV$_2$+(I+III$_2$+II)$_2$, suggesting the functional restoration of the electron pathway from the distal Q$_P$ site to the proximal Q$_N$ site.

Functional loss of the distal Tt-CIII$_2$ protomer in Tt-SC I+III$_2$ and its restoration in Tt-SC IV$_2$+I+III$_2$+II and Tt-MC IV$_2$+(I+III$_2$+II)$_2$ lead to a tempting possibility that the distal Q$_P$ site is specialized in oxidizing QH$_2$ generated by Tt-CII. Structurally, a notable cleft between ND5a's bulk portion and its Tt-specific CT TMH constitutes a hydrophobic channel connecting two lipid-filled cavities separated by Tt-CI MA toe, reaching respectively to Tt-CII's Q reduction cavity and to the distal Q cavity of Tt-CIII$_2$ (Fig. 3a and Supplementary Fig. 14d). Several lipid molecules fill this cleft, indicating enough space for the less bulky QH$_2$ with a single hydrocarbon tail to get through (Fig. 3a). It is noteworthy that the long horizontal helix at mammalian ND5 CT is split into an independent ND5b subunit in *T. thermophila*, making the existence of an additional Tt-specific ND5a CT TMH and the connective channel possible (Supplementary Fig. 14d). Moreover, Q density was found on a proximal position near the UQCRTT2 TMH and subunit COB TMH7-8, but was absent on the same position of the distal side in Tt-SC I+III$_2$[3]. This non-canonical Q site might represent a position of local binding energy minimum where QH$_2$ is to leave the bulk membrane phase and access the proximal Q$_P$ site through the cleft between TMHs of

subunits COB and UQCRQ (Supplementary Fig. 14c). The idea of such en route Q site has been proposed for mammalian CI as the shallow Q site near its Q tunnel entrance[63,64] and as the leaving site for menaquinone next to the CIII$_2$ Q$_P$ site in the actinobacterial SC III$_2$+IV$_2$[65]. In Tt-MC IV$_2$+(I+III$_2$+II)$_2$, equally clear Q densities occupy both en route Q sites of Tt-CIII$_2$, implying QH$_2$ supply to not only the proximal but also the distal Q$_P$ sites (Fig. 3b and Supplementary Fig. 12j, k).

It is noteworthy that succinate alone can fuel oxidative phosphorylation of *T. thermophila* mitochondria[33], demonstrating the physiological significance of CII incorporation into megacomplex as an alternative QH$_2$ source compared to mammalian mitochondria[66]. On the other hand, whether a channeling Q pool sealed inside mammalian respirasome and megacomplex is segmented from the other freely diffusing Q/QH$_2$ in the bulk membrane phase, is currently under debate[67–69]. In *T. thermophila* ETC, the incorporation of Tt-CII into megacomplex, the existence of the hydrophobic channel at ND5a CT and the crowdedness in IMM due to possible stacks of ETC rings architecture support the idea of QH$_2$ channeling between Tt-CII and the distal Q$_P$ site of Tt-CIII$_2$. Kinetically, the NADH:O$_2$ and succinate:O$_2$ pathways seem to be mutually inhibitory when operating simultaneously, as suppressing Tt-CII activity by adding increasing doses of malonate stimulates Tt-CI activity (Supplementary Fig. 3a, b). This argues against the idea of QH$_2$ channeling and is in favor of a shared Q pool between the two pathways, but can alternatively be due to the two pathways competing for a shared cyt c pool externally added to make the activities measurable[67]. Whether Tt-CIII$_2$'s two protomers are specialized in separately receiving electrons from Tt-CI and Tt-CII in megacomplexes and whether such mechanism regulates *T. thermophila* ETC turnover and higher-order assembly in adaptation to its lifestyle of constant swimming remain to be explored experimentally[70].

## Discussion
Our findings demonstrate the structural divergence of *T. thermophila* ETC megacomplex in adaptation to the tubular morphology of its mitochondrial cristae. Although the highly diversified Tt-CIV$_2$ leads to its separation from Tt-CIII$_2$ in a single Tt-MC IV$_2$+(I+III$_2$+II)$_2$, it contacts Tt-CI MA toe in the neighboring megacomplex so that spatial proximity to Tt-CIII$_2$ is maintained, likely to facilitate cyt c diffusion. Nonetheless, Tt-MC (IV$_2$+I+III$_2$+II)$_2$ only represents a minor assembly species of digitonin-extracted *T. thermophila* IMM as demonstrated by its particle number during cryo-EM image processing (Supplementary Fig. 6), implying structural instability at the interface between two Tt-SC IV$_2$+I+III$_2$+II. This is in line with the highly plastic cristae tube morphology as revealed by cryo-electron tomographic (Cryo-ET) studies[30], as different degrees of tubular curvatures can only be accommodated by flexible interfaces among stacks of ring-like ETC assemblies. Therefore, stable assembly with direct CIII$_2$-CIV contact must be lacking in *T. thermophila* IMM, which is in stark contrast to opisthokonts and archaeplastidans with isolatable SC III$_2$+IV$_{1/2}$[4–7]. Physiological significances of SC III$_2$+IV$_{1/2}$ formation include reducing the cyt c diffusion distance and kinetically optimize electron transfer between the two complexes[24]. Moreover, the concept of 2D diffusion has been raised for yeast SC III$_2$+IV$_{1/2}$ as a negatively charged patch between its CIII$_2$ and CIV cyt c sites serves as a sliding track for the positively charged cyt c channeling in between without equilibration with the bulk IMS phase[6]. Although no such charged surface patch has been identified for Tt-MC IV$_2$+(I+III$_2$+II)$_2$ and Tt-MC (IV$_2$+I+III$_2$+II)$_2$, the nearest distance (~97 Å) between cyt c reduction and oxidation sites in Tt-MC (IV$_2$+I+III$_2$+II)$_2$ is comparable to that in a mammal (85 Å)[4], yeast (61 Å)[7], and plant (70 Å)[5]. The current organization of Tt-SC IV$_2$+I+III$_2$+II separating Tt-CIII$_2$ from Tt-CIV is likely due to other structural and functional requirements. The large, obligatory dimeric Tt-CIV$_2$ obviously requires a more extensive association interface than the narrow Tt-CI MA toe, while the incorporation

of Tt-CII into the megacomplex necessitates spatial adjacencies of both Tt-CI and Tt-CII to Tt-CIII$_2$ to reduce the diffusion distances of QH$_2$. The increased distance between Tt-CIII$_2$ and Tt-CIV sites within Tt-MC IV$_2$ + (I + III$_2$ + II)$_2$ is compensated by the reversible contact between Tt-MC IV$_2$ + (I + III$_2$ + II)$_2$ assemblies, providing an overall evolutionary advantage to *T. thermophila* ETC.

Supercomplex assembly factor 1 (SCAF1)[4,71], hypoxia-inducible gene domain (HIGD)[72,73], and NDUA4[74,75] have been proposed as assembly factors in the ETC complex and supercomplex maturation of Opisthokonta. Among these, NDUA4 has not been detected in mammalian CI, CIV, or CIV$_2$ except for the human[36] and, to a lesser extent, ovine respirasomes[8,76]. It has neither been proposed as a CI-CIV linker due to its non-interfacial position in the respirasome[76], nor as a CIV dimerization factor considering the potential steric clashes it would introduce at the dimer interface[38]. In the ciliate respiratory chain, however, its proposed role inhibiting CIV dimerization can be ruled out due to its presence on the dimer interface of Tt-CIV$_2$ (Supplementary Fig. 13a). Such functional distinction of the same assembly factor reflects its divergent evolution after branching between Opisthokonta and Alveolata from the Last Eukaryotic Common Ancestor (LECA). Apart from NDUA4, SCAF1 has been found only in mammalian SC III$_2$ + IV[4] but not respirasome SC I + III$_2$ + IV[8,9]. HIGD1A and HIGD2A have not been observed in any mammalian SC structure, but their yeast isoform Rcf2 is present in its SC III$_2$ + IV$_2$ under hypoxic conditions[36]. COXTT17 structurally superposes with Rcf2. However lack of the conserved QRRQ sequence motif[72] and the aerobic culturing condition of *T. thermophila* argue against its identity as a HIGD family member (Supplementary Fig. 13b). COX7A, the structural homolog of SCAF1 endogenous in Opisthokont CIV[77], is also present in Tt-CIV$_2$ (Supplementary Fig. 13c) but either serve as place holders for homologous assembly factors or have evolved into pure structural subunits in the Alveolata clade.

A major difference between mammalian MC (I + III$_2$ + IV)$_2$ and Tt-MC IV$_2$ + (I + III$_2$ + II)$_2$, apart from the incorporation of Tt-CII, is that the central coupler linking two CI copies together are mammalian CIII$_2$ and Tt-CIV$_2$, respectively. The half ring-shaped architecture of Tt-MC IV$_2$ + (I + III$_2$ + II)$_2$ explicitly indicates that ETC complexes are packed into stacks of rings on *T. thermophila* tubular cristae. Questions are raised as to whether *T. thermophila* ETC is further organized into a full ring and what possible complex arrangement the full ring could have. A simple concatenation of two Tt-MC IV$_2$ + (I + III$_2$ + II)$_2$ by the obligatory dimeric Tt-CIII$_2$, as in mammalian and plant megacomplexes, would extend the ETC assembly curvature to 320°. However, it is unlikely to happen since superposing two Tt-MC IV$_2$ + (I + III$_2$ + II)$_2$ by Tt-CIII$_2$ introduces an elliptical instead of circular assembly (Supplementary Fig. 15). Significant conformational plasticity within Tt-SC I + III$_2$ would be required if such organization are to take place on the tubular cristae. Meanwhile, the existence of a Tt-CV$_2$ row along its tubular cristae, as revealed by Cryo-ET studies[30], indicates that Tt-CV must also be incorporated into the possible respiratory rings. The symmetric structure of Tt-CV$_4$ suggests an association of similar protein complexes on both sides when viewed axially, as the situation of Tt-CIV$_2$ within Tt-MC IV$_2$ + (I + III$_2$ + II)$_2$. Whether such protein complexes adjacent to Tt-CV$_4$ is one of the four ETC complexes or involve, unidentified structural elements is currently unknown. Therefore, the full architecture of the respiratory ring, the way Tt-CV$_4$ is incorporated into such ring and possible variabilities in spatial register between neighboring rings await future cryo-electron microscopic and cryo-tomographic investigations.

## Methods

### Cell culture and mitochondrial purification

Wild-type *T. thermophila* SB210 was provided by the National Aquatic Biological Resource Center, NABRC, under the catalog number NABRC.NCCAP.IHBPTTHE000248 and was cultured in SSP media (2% protease peptone, 0.1% yeast extract, 0.2% glucose, and 33 μM FeCl$_3$) at 35 °C and 125 rpm shaking until $1 \times 10^6$ cells/ml density. Harvested cell pellets were lysed manually in TMIBS buffer (10 mM Tris pH 7.4, 2 mM EDTA, 0.3 M sucrose, 0.002% PMSF (w/v)) with a pre-cooled KIMBLE Dounce tissue grinder. To get rid of the exocytosed mucocyst, a crude mitochondrial fraction of the lysate was washed twice in TMIBS buffer with 10% percoll (Yaseen Biotechnology) before pelleting down by centrifugation at 7000 × g for 10 min. This crude mitochondria was further purified on a discontinuous sucrose gradient with 30, 45, and 60% sucrose (w/v) in TMIB buffer (10 mM Tris pH 7.4, 2 mM EDTA, and 0.002% PMSF) by centrifugation at 140,000 × g for 2 h in a SW32 Ti rotor (Beckman Coulter). Fine mitochondria fraction was harvested from the 45–60% sucrose interface and pelleted by final centrifugation at 16,000 × g for 45 min. The mitochondria pellet was weighed and stored at −80 °C until use.

### Electron transport chain megacomplexes purification

The following steps were performed at 4 °C unless otherwise indicated. To isolate the mitochondrial membrane fraction from *T. thermophila* mitochondria, thawed mitochondria pellets were homogenized in 10 ml/g (v/w of the mitochondrial pellet) pure water manually in a pre-cooled KIMBLE Dounce tissue grinder. About 3 M KCl was then added to the homogenate to a final KCl concentration of 150 mM, followed by brief manual homogenization. The homogenate was then centrifuged at 32,000×g for 45 min to pellet down the membrane fraction. The pellet was resuspended in 18 ml/g (v/w of membrane pellet) M10 buffer (20 mM Tris pH 7.4, 50 mM NaCl, 1 mM EDTA, 2 mM DTT, 10% glycerol, and 0.002% PMSF (w/v)), followed by centrifugation at 32,000×g for 45 min. The pellet was again resuspended in 3 ml/g (v/w of membrane pellet) M10 buffer. The total protein concentration of the resuspended membrane fraction was determined by Bradford colorimetric assay (BCA), then added by M90 buffer (20 mM Tris pH 7.4, 50 mM NaCl, 1 mM EDTA, 2 mM DTT, 90% glycerol, and 0.002% PMSF (w/v)) to final glycerol concentration of 30%. The mitochondrial membrane fraction was stored at −80 °C until use.

For Tt-MC IV$_2$ + (I + III$_2$ + II)$_2$ purification, thawed mitochondrial membrane (-55 mg of total protein) was solubilized by slow rotating for 1 h in buffer M10 (20 mM Tris pH 7.4, 50 mM NaCl, 1 mM EDTA, 2 mM dithiothreitol (DTT), 0.002% PMSF (w/v), 10% glycerol (v/v)) with 1% lauryl maltose neopentyl glycol (LMNG) (w/v) at a detergent-to-protein ratio of 2:1 (w/w), before centrifugation at 16,000×g for 45 min. The supernatant was concentrated to -3 ml in a centrifugal concentrator with 100 KDa molecular weight cut off (MWCO) (Millipore), which was used for all the following concentration steps in this section. The concentrated sample was loaded onto a continuous 20–50% (w/v) sucrose gradient in SGB buffer (15 mM HEPES pH 7.7, 20 mM KCl, and 5 mM MgCl$_2$) with 0.005% LMNG (w/v) by 140,000×g centrifugation for 20 h using a SW32 Ti rotor (Beckman Coulter). Gradients were fractionated manually into 1 ml fractions, before running them on 2–8% BN-PAGE and in-gel CI activity nitrotetrazoleum blue assay. Fractions containing the two upper bands corresponding to Tt-MC IV$_2$ + (I + III$_2$ + II)$_2$ and IV$_2$ + I + III$_2$ + II were pooled, concentrated, and loaded onto a Superose 6 Increase 10/300 GL column (GE Healthcare) for size-exclusion chromatography in SEC buffer (20 mM Tris pH 7.4, 50 mM NaCl, 0.002% PMSF, and 0.005% LMNG (w/v)). Tt-MC IV$_2$ + (I + III$_2$ + II)$_2$-enriched fraction eluting at -9 ml was concentrated to 4 mg/ml and used immediately for cryo-EM grid preparation.

Tt-SC (IV$_2$ + I + III$_2$ + II)$_2$ was purified likewise. Briefly, the thawed mitochondrial membrane (-55 mg of total protein) was solubilized by slow rotating for 1 h in buffer MX (30 mM HEPES pH 7.7, 150 mM potassium acetate, 0.002% PMSF, 10% (v/v) glycerol) with 2% digitonin at a detergent-to-protein ratio of 5:1 (w/w). After centrifugation at 140,000 × g for 20 h in a continuous 20–50% (w/v) sucrose gradient in SGB buffer with 0.1% digitonin (w/v), fractions at 33–38% sucrose

concentration were pooled, concentrated, and loaded onto the Superose 6 Increase 10/300 GL column equilibrated in SEC buffer with 0.1% digitonin (w/v). The fraction eluted at 8–8.5 ml was diluted by SEC buffer with 0.1% digitonin (w/v) to ~1.5 mg/ml and used immediately for cryo-EM grid preparation.

## Tt-cytochrome c (cyt c) expression

pET22b(+) plasmid containing synthesized Tt-cyt $c$ gene (UniProt: I7MFH3) immediately between the pelB periplasmic signal sequence and the 6×His tag was transformed into *E. coli* BL21 strain by Tsingke Biotechnology. Plasmid pEC86 containing the *E. coli* ccmABCDEFGH operon essential for maturation of cyt $c$ was purchased from the Culture Collection of Switzerland[78] and transformed into the *E. coli* BL21 strain carrying the pET22b(+) plasmid. A single double-transformed colony was cultured in 15 ml of LB containing 100 µg/ml ampicillin and 34 µg/ml chloramphenicol at 37 °C with 250 rpm shaking for about 12 h. Then 100 ml LB with the same antibiotics was inoculated with 10 ml of this start culture and cultured similarly overnight. The next morning, eight 1 L conical flasks, each containing 500 ml LB with the same antibiotics, were each inoculated with 10 ml of the previous culture and cultured at 30 °C with 250 rpm shaking for about 4 h until the optical density at 600 nm reached between 0.9 to 1. The cultures were then induced with 30 µM IPTG and cultured at 30 °C with 190 rpm shaking overnight.

The next morning, the bacterial cells were harvested by centrifugation for 15 min at 3000×$g$ at 4 °C, then resuspended in ice-cold TES buffer (100 mM Tris pH 7.8, 20% sucrose w/v, and 0.5 mM EDTA) containing 0.002% PMSF (30 ml TES buffer per pellet from 1 L of culture). The resuspensions were transferred into 300 ml bottles, added with 0.5 mg/ml lysozyme, and incubated for 15 min at room temperature. The bottles were then placed on an ice tray, added with equal volumes of ice-cold ultra-pure water containing 0.002% PMSF and bovine DNase A, then rocked at 90 rpm for 30 min. The suspension was then centrifuged at 12,000×$g$ for 30 min at 4 °C, the supernatant of which was transferred to fresh tubes and filtered through a 0.45 µm filter. The cleared supernatant was loaded onto a 5 ml SP Sepharose FF column and eluted with a 0−500 mM NaCl gradient over 10 column volumes at a flow rate of 1 ml/min. Peak fractions monitored by absorbance at 415 and 550 nm were pooled and concentrated with 3 KDa MWCO centrifugal concentrators (Millipore) to a final concentration of ~9.7 mg/ml (~0.8 mM), estimated by BCA assay.

The presence and size of Tt-cyt $c$ were determined by SDS-PAGE with a 12% Tris-SDS gel at 120 V before staining with Coomassie brilliant blue for 20 min (Supplementary Fig. 2a). To measure the extinction coefficient of Tt-cyt $c$, 0.06% $H_2O_2$ (v/v) or 4 mM DDT were used to fully oxidize or reduce the Tt-cyt $c$. The redox state of the purified Tt-cyt $c$ were measured by a spectral scan from 200 to 800 nm in 1 nm increments using a Thermo Scientific Spectrophotometer, from which an $A_{550\,nm}/A_{565\,nm}$ ratio >9.0 was considered fully reduced (Supplementary Fig. 2b). Standard curves for reduced and oxidized Tt-cyt $c$ were created by measuring absorbance at 550 nm for 26.8, 53.6, 80.4, 107.2, and 134 µM dilution of the 0.8 mM Tt-cyt $c$ stock done in three replicates. Values in the standard curves are averages ± SEM. The reduced and oxidized extinction coefficients of Tt-cyt $c$ were calculated by dividing the slopes of the two standard curves by the optical path length (0.2 cm), giving a reduced−oxidized Tt-cyt $c$ extinction coefficient of 6.36 mM$^{-1}$cm$^{-1}$ (Supplementary Fig. 2c).

## Spectroscopic assays for megacomplex $IV_2 + (I + III_2 + II)_2$ activity

Tt-MC $IV_2 + (I + III_2 + II)_2$ activity was determined by spectroscopic observation of individual and simultaneous NADH and succinate oxidation kinetics in the absence and presence of CI (rotenone), CII (malonate), CIII$_2$ (antimycin A), and CIV (sodium azide) inhibitors at 340 and 550 nm. NADH (Aladdin), succinate (Sigma-Aldrich),

decylubiquinone (DQ; Aladdin), rotenone (Sigma-Aldrich), malonate (Aladdin), antimycin A (MKbio), and sodium azide (Sigma-Aldrich) were used for following activity measurements as needed. All activities were measured in 384-well plates at room temperature using a Thermo Scientific Spectrophotometer with a total reaction volume of 20 µl per well. Extinction coefficients of 6.22 and 4.52 mM$^{-1}$cm$^{-1}$ was used for NADH and Tt-cyt $c$ in the activity calculations.

The reaction master mix consisted of 20 mM Tris-HCl pH 7.4, 50 mM NaCl, 0.1% digitonin (w/v), 0.002% PMSF, 50 µM decylubiquinone (DQ), and 100 µM Tt-cyt $c$. For NADH:$O_2$ and succinate:$O_2$ oxidoreductase activities of Tt-MC $IV_2 + (I + III_2 + II)_2$, 10 nM Tt-MC $IV_2 + (I + III_2 + II)_2$ was used and where indicated, the relevant respiratory inhibitors (50 µM rotenone, 5 mM malonate, 500 µM or 1 mM antimycin A, or 100 mM NaN$_3$) were added. The reaction was initiated by the addition of 500 µM NADH or 5 mM succinate, mixed in the spectrophotometer for 5 s, then recorded every 6 s at 340 and 550 nm simultaneously. The NADH:$O_2$ and succinate:$O_2$ oxidoreductase activities in Fig. 1d, e, g were calculated as NADH oxidation and cyt $c$ reduction, based on the slopes of the initial linear phases between 50 and 150 s in the 340 and 550 nm kinetic curves, respectively (Supplementary Fig. 3c, d). The NADH:$O_2$ and succinate:$O_2$ oxidoreductase activities in Fig. 1f, h were calculated as cyt $c$ oxidation, based on the slopes of the last decline phases (1200s–1500s for the control and sodium azide curves, 3000s–3300s for the blank curve in Supplementary Fig. 3c right panel; 5200s–5500s for the control, sodium azide and blank curves in Supplementary Fig. 3d). Note that CIV inhibitor sodium azide could not inhibit cyt $c$ reduction by Tt-CIII$_2$ but can inhibit its oxidation by Tt-CIV$_2$ in both NADH:$O_2$ and succinate:$O_2$ oxidoreductase activity measurements.

For comparison of the activities of individual and simultaneous NADH and succinate oxidation in the absence and presence of increasing doses of malonate by the purified Tt-MC $IV_2 + (I + III_2 + II)_2$, 2 nM Tt-MC $IV_2 + (I + III_2 + II)_2$ were used for the activity assay and where indicated, 50, 100, 500 µM, or 5 mM malonate was added to the reaction mixture (Supplementary Fig. 3a, b). The reaction was initiated by the addition of 250 µM NADH and/or 10 mM succinate, mixed in the spectrophotometer for 5 s, then recorded every 6 s at 340 and 550 nm simultaneously. The NADH:$O_2$ and succinate:$O_2$ oxidoreductase activities were calculated as NADH oxidation and cyt $c$ reduction, based on the slopes of the initial linear phase between 50 and 300 s in the 340 and 550 nm kinetic curves, respectively.

## Cryo-EM grid preparation and data collection

The cryo-EM grid preparation and data collection were carried out at the Center of Cryo-Electron Microscopy at Zhejiang University. For Tt-MC $IV_2 + (I + III_2 + II)_2$ (Dataset 1), a 3 µl sample was applied to Quantifoil 0.6/1 copper grids freshly glow discharged at 25 mA for 120 s, blotted by Vitrobot Mark IV (Thermo Fisher) for 8 s under 100% humidity at 4 °C and plunged-frozen by liquid ethane. For Tt-SC $(IV_2 + I + III_2 + II)_2$ (Dataset 2), a 3 µl sample was applied to a Quantifoil R1.2/1.3 300 mesh copper grid with 2 nm thin layer of carbon freshly pre-glow discharged at 15 mA for 15 s. The blotting procedure is the same as for Dataset 1. Grids were transferred to liquid nitrogen before screening under a 200 kV Talos Arctica microscope (Thermo Fisher).

Grids with correct particle shape, density, and dispersion were used for the data collection on a Titan Krios microscope (Thermo Fisher) operated at 300 kV equipped with a Falcon4 detector operating at 241 frames/s and a Selectris energy filter. Automated data collection was performed using EPU software following standard procedures with a defocus range from −0.8 to −2.0 µm. For Dataset 1, the grids were imaged at a calibrated magnification of ×140,000 with a physical pixel size of 0.93 Å. A total dose of 61.5 e$^-$/Å$^2$ with 7 s exposure time was fractionated into 1687 frames and a total of 16,772 movies were collected. For Dataset 2, the grid was imaged at a calibrated magnification of ×105,000 with a physical pixel size of 1.2 Å on images.

A total dose of 51.4 e⁻/Å² with 8.46 s exposure time was fractionated into 2039 frames and a total of 11,182 movies were collected.

## Cryo-EM image processing

For both datasets, motion-correction was performed using Relion-4.0's own implementation[79], before contrast transfer function (ctf) estimation using CTFFIND4.1[80]. Particles were picked in crYOLO[81] and extracted in Relion-4.0 using box sizes 840 for Dataset 1 and 700 for Dataset 2. The following steps were performed in cryoSPARC v3.3.2[82]. For Dataset 1, a total of 434,449 particles were generated after particle curation by 2D classification and 3D ab-initio reconstruction. Overall homogeneous refinement with global and local ctf corrections and C1 symmetry gave a resolution of 2.89 Å. Focused 3D classification on the Tt-SC $I + III_2 + II$ region that was absent in Tt-SC $IV_2 + I + III_2 + II$ separated 97,688 Tt-MC $IV_2 + (I + III_2 + II)_2$ particles from 336,761 Tt-SC $IV_2 + I + III_2 + II$ particles. Overall non-uniform refinement[83] with global and local ctf corrections and C2 symmetry of the Tt-MC $IV_2 + (I + III_2 + II)_2$ particles gave a resolution of 2.96 Å. After re-extraction centered on each Tt-SC $IV + I + III_2 + II$ protomer, thereby doubling the particle number, local refinements were performed on Tt-CIV, Tt-CI PA, Tt-MA proximal region, Tt-MA distal region, Tt-CIII₂ and Tt-CII, giving resolutions of 2.89, 2.96, 2.80, 2.83, 2.86, and 3.26 Å. These regional maps were combined into a composite map using the Combine Focused Maps program in Phenix-1.20.1[84]. For Dataset 2, a total of 325,256 particles were generated after particle curation by 2D classification and 3D ab-initio reconstruction. Overall homogeneous refinement with C1 symmetry gave a resolution of 2.89 Å. Focused 3D classification on the Tt-SC $IV_2 + I + III_2 + II$ region that was absent in Tt-SC $IV_2 + I + III_2 + II$ and Tt-MC $IV_2 + (I + III_2 + II)_2$ gave 19,023 Tt-SC $(IV_2 + I + III_2 + II)_2$ particles, overall homogeneous refinement of which gave a resolution of 4.18 Å, but only one copy of Tt-SC $IV_2 + I + III_2 + II$ had sensible density features. Local refinement of the second copy of Tt-SC $IV_2 + I + III_2 + II$ gave a resolution of 6.77 Å, which was combined with the former map into a composite map in Phenix-1.20.1. The composite maps from both datasets were used for the following automatic and rigid-body model refinements.

## Model building and refinement

For Dataset 1, initial models for Tt-CIV₂ (PDB: 7W5Z) and Tt-SC $I + III_2$ (PDB: 7TGH)[3] were rigid-body fit into the composite map of Tt-MC $IV_2 + (I + III_2 + II)_2$ in ChimeraX[85]. For Tt-CII, core subunits, including SDHA, SDHB, SDHC, and SDHD were homology modeled using the Phyre2 server[86] and rigid-body fit into the Tt-CII region of the composite map. All manual model building was performed in Coot-0.9.6[87]. Ciliate-specific subunits were first built as a poly-alanine model, side chains were then added de novo according to density. The generated query sequences were used in BLAST searches against the *T. thermophila* UP000009168 database from UniProt. Secondary structure information predicted by the PredictProtein server[88] was also used to assist modeling in Coot-0.9.6. The manually built Tt-SC $IV + I + III_2 + II$ model was rigid-body fit into both Tt-SC $IV + I + III_2 + II$ protomers of the composite Tt-MC $IV_2 + (I + III_2 + II)_2$ map, before being combined into a Tt-MC $IV_2 + (I + III_2 + II)_2$ model with chain ID adjustments. Automatic model refinement was performed using the phenix.refine and phenix.real_space_refine programs against the composite Tt-MC $IV_2 + (I + III_2 + II)_2$ map. We automatically generated secondary structure restraints, custom bond linkage, and custom ligand description files by Phenix-1.20.1 before manually editing according to the outcome of the automatic refinement. The refined model was manually edited in Coot-0.9.6 before the next round of automatic refinement until the refined model achieved high model-map correlation and good geometric statistics. A total of 326 subunits were built for Tt-MC $IV_2 + (I + III_2 + II)_2$, among which 11 ciliate-specific subunits were uniquely identified for Tt-CII. For Dataset 2, individual Tt-CI, Tt-CIII₂, and Tt-CIV₂ models from the refined Tt-SC $IV_2 + (I + III_2 + CII)_2$ model

was rigid-body fit into Tt-SC $(IV_2 + I + III_2 + CII)_2$ map and combined without further refinement.

## Reporting summary

Further information on research design is available in the Nature Portfolio Reporting Summary linked to this article.

## Data availability

The structural models of Tt-MC $IV_2 + (I + III_2 + II)_2$ and Tt-MC $(IV_2 + I + III_2 + CII)_2$ generated in this study have been deposited in the Protein Data Bank (PDB) under accession codes 8GYM and 8GZU respectively. The composite Cryo-EM maps of Tt-MC $IV_2 + (I + III_2 + II)_2$ and Tt-MC $(IV_2 + I + III_2 + CII)_2$ generated in this study have been deposited in the Electron Microscopy Database (EMDB) under accession codes EMD-34373 and EMD-34403. Local refinements of CI peripheral arm, CI membrane arm distal region, CI membrane arm proximal region, CIII₂, CIV, and CII of Tt-MC $IV_2 + (I + III_2 + II)_2$ generated in this study have been deposited in the EMDB under accession codes EMD-34380, EMD-34381, EMD-34382, EMD-34384, EMD-34383, and EMD-34385 respectively. Local refinements of the two Tt-SC $IV_2 + I + III_2 + CII$ regions of Tt-MC $(IV_2 + I + III_2 + CII)_2$ generated in this study have been deposited in the EMDB under accession codes EMD-34404 and EMD-34405 respectively. The structural models used in this study are available in the PDB under accession codes 5XTI, 1ZOY, 7W5Z, 7TGH, 5IY5, 3CX5, 5J4Z, 6QBX, and 8BPX. Source data are provided with this paper.

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

## Acknowledgements

We thank SH. Chang and LY. Wu in the Centre of Cryo-Electron Microscopy (CCEM), Zhejiang University, for their technical assistance on Cryo-EM grid screening and data collection. We thank C. Ma and LY. Wang from the Protein facility, Core facilities, and Zhejiang University School of Medicine for their assistance. We are grateful to Dr. S. Zhang from the Department of Biochemistry, School of Medicine, Zhejiang University, for the critical reading of the manuscript. This project is supported by the ZJU100 Young Professor Award from the School of Medicine, Zhejiang University.

## Author contributions

Conceptualization, methodology, funding acquisition, project administration, and supervision: L.Z.; Investigation: L.Z., F.H., Y.H., M.W., Z.H., and H.T.; Visualization: L.Z., Y.H., and M.W.; Writing—original draft and Writing—review and editing: L.Z., Y.H., Z.H., and M.W.

## Competing interests

The authors declare no competing interests.
