## [Peer review file · Nature Communications]

REVIEWER COMMENTS

Reviewer #1 (Remarks to the Author):

The manuscript by Han et al. reports cryo-EM structures of a respiratory “megacomplex”. The manuscript is not easy to follow and it is rather descriptive. Many terms are incorrectly used. The authors do not present any activity measurements of the components of the "megacomplex". Such measurements must be included, otherwise the work shows only a series of structures, but we don't know if these structures represent active enzymes. Is there a "megacomplex" activity?

Specific comments:

L28 What is "oxidative energy"?

L30 "proton motive force"

L76 What is "conformational rigidity". The term must be defined before use.

Fig. 1 legend. The authors refer to "diffusion direction". There is no diffusion direction for cyt c. It diffuses randomly in the water phase. Do the arrows show a sequence of reactions?

L123 Differences between the current structure and that from mammalian mitochondria are discussed and attributed to "highly divergent Tt-CIV2". Details of these differences must be discussed, otherwise the statement has no meaning.

L134- Distances for transfer of cyt c are discussed, but it is not clear why. The discussion is descriptive. The authors must elaborate on the consequences of the measured distances on the functionality.

L177-79 Difficult to understand. What is "WT", "electron capacitor"?...

L186-190 The authors speculate on "structural role" vs. participation in electron transfer. The activity of the complex (electron transfer) can be measured. The authors should do the measurement rather than presenting speculations from which the reader cannot draw any conclusions.

L217-230 The discussion doesn't make sense. Complex III transfers protons across the membrane by a Q-cycle mechanism. The statements here suggest that this is not the case. Please clarify. This section is also difficult to read and understand so I may have misunderstood the conclusions.

Complexes II and III are presented in the main text, but a discussion on complex IV is missing. Please add.

Reviewer #2 (Remarks to the Author):

Han et al. present a structural study of the mitochondrial respiratory megacomplexes of the ciliate model organism *Tetrahymena thermophila*. In their study, the authors reveal the architectures of two further megacomplexes that adapt to the curvature of the cristae membrane. This is a crucial question that has remained elusive for years in what way the respiratory megacomplexes affect mitochondrial morphology and the mechanism that results from the composition and architecture of these molecular machines. In this manuscript, the authors present ~8 MDa megacomplex $IV_2+(I+III_2+II)_2$, as well as a ~10.6 MDa megacomplex $(IV_2+I+III_2+II)$. Both complexes reveal different structural rearrangements. In the first structure CIV2 associates one copy of supercomplex I+III₂ and one copy of CII to shape the cristae membrane curvature. The other arrangement aims the proximity between CIV2 and CIII₂ cytochrome c binding sites. Taking together, the authors' analysis defines new details of the divergence

in eukaryotic electron transport chain organization and how it is related to mitochondria morphology and function.

The manuscript is well written and gives an expanded view of existing studies from super- and megacomplexes regarding their architectures, function and evolution. References are sufficient and SI figures are all necessary to support the current manuscript. I recommend to publish the manuscript in Nature Communication. However, I have only some minor points.

- 1) Labels need to be better explained in the figure legends. For example, at Fig. 3C. What is SC102 and SB 404? Because it is not listed in the main text either.
- 2) The authors should include scale bars for the raw micrographs in sFig 3.
- 3) Would it be possible to show the angular distribution of particles at least for the two megacomplexes in sFig 4?
- 4) The authors should present side chains in sFig 6 M & N.
- 5) In the legend of sFig. 9 “Tt-CI” has to be changed into “Tt-CII”.
- 6) Are the ratios of the mammalian and ciliate supercomplexes in sFig. 10 A & B correct? It is also difficult to differentiate complexes I - IV.
- 7) The publication from Mühleip et al. (2022) should already be cited in the introduction as common points are addressed in the manuscript.
- 8) I found no references to Supplementary Tables 1 and 2 in the text.

Reviewer #3 (Remarks to the Author):

This manuscript describes a Cryo-EM description of an approximately 8 and 10.6 MDa supercomplex of the mitochondrial respiratory chain from the ciliate *Tetrahymena thermophila*. The scope of the work is very impressive, however, the manuscript seems to have been hastily put together and it would seem a significant revision is necessary to make it more accessible to the reader. In mammals the mitochondrial respiratory supercomplex is usually composed of complexes I, III, and IV. In humans, however, it has been modeled that a large gap between complexes I and IV may allow complex II to insert into the supercomplex (this is modeled in Ref. 10). The current study clearly supports the notion that in *T. thermophila* that complex II is part of these megacomplexes. Additionally in *T. thermophila* many additional subunits are associated with the respiratory complexes such that 364 individual subunits (compared to about approximately 75 individual subunits in mammalian supercomplexes) have been identified defining the entire electron transport chain of *T. thermophila*. It is curious that the authors do not provide any suggestions for why these many additional subunits are needed for the respiratory chain of *T. thermophila*. A brief discussion would be useful. It is realized that there is likely no definitive answer for why there are so many additional subunits in the single cell ciliate, however reasonable speculation would be appreciated. It should be noted that there is some discussion by the authors regarding the potential role of certain ciliate specific subunits in the Discussion (lines 334-359) regarding the relationship to mammalian supercomplex assembly factors. This sort of information is useful but it seems like it is really part of the Results and not necessarily for the Discussion.

Overall the manuscript provides very interesting information about the ciliate respiratory supercomplex but a more detailed analysis of the data and data analysis would make this a much more useful contribution.

Specific comments: These are listed in order as they appear in the text but some are much more significant issues than others.

1. Page 2, line 54. In Supplementary Fig. 1, line 27; do the authors mean the pooled fractions from B and F or do they mean from A and E. The green boxes indicating the samples used for Cryo-EM appear in B and F, but also the blue hatching in A and E seem to indicate specific fractions although that just may refer to the gel(s) below.
2. Also regarding Fig. S1. Although the authors clearly show in their structure that complex II is part of the “megacomplex” none of the bands in Fig. S1, B, D, F, or H ever indicate the presence of complex II. It must be somewhere in the gel? One assumes that the samples labeled Ti-MC or Ti-Sc I, III, IV must have also included complex II? Is that the case? Please clarify.
3. Supplementary Fig. 2 legend. Please define what the pinkish region is supposed to represent since the same color seems to be used to represent different complexes and different regions of the same complex.
4. Line 73. The authors begin discussing Supplementary Fig. S4 before they discuss Supplementary Figure S3. Why not swap Figs. S3 and S4? They should re-order and renumber their figures so that the

reader doesn't have to hunt around for the figure. It is not always clear which figure the authors are referring to or if they have made a typo in the text.

5. Line 83 and 84. The authors bring up the idea that membrane arm (heel) of complex I is opposite the side of the ubiquinone (Q) tunnel. Since they have never discussed the Q tunnel to this point of the manuscript this is hard to follow. Also no Q-tunnel is shown in Fig. 1 so maybe this comment is premature or the phrasing needs to be altered.

6. The colors chosen to represent CII and CIV in Fig. 1A are quite similar and makes it hard to distinguish the subunits. The bolder colors in Fig. 1B (dark red for CII, magenta for CIV) for these two complexes might make a much sharper figure where it is easier to discern the boundaries between these complexes. The same comment can be made about Fig. 2.

7. Line 85. It is actually Ref. 10 (not 24 as indicated in the text) where the "wedge" between CI and CIV is suggested as a site for CII to bind.

8. Line 97. Once again a supplementary figure is discussed out of order. Fig. S7 before any mention of Fig. S6. Fix this.

9. Line 104. Fig. S10 C and D discussed before any mention of Figs. S6, 8 or 9.

10. Line 100. Please confirm that Reference 15 is the proper reference for the term "respiratory patch".

11. Line 118. The authors need to refer to their Supplementary Table 2 when they begin to discuss the individual subunits of the complexes and especially when referring to the many additional subunits from the ciliate.

12. Table S2. It would be useful to indicate a reference for the assignment of the role of the proteins listed in Table S2. For example, in Line 118 it is stated that COXMC3 is a mitochondrial carrier subunit. How/who determined this? If is just how it is annotated in databases please indicate which database was used.

13. Line 119 and 120. It is not obvious from Fig. 2A that COXMC1 interacts with CII. Again the color contrast in the figure makes this point very obscure.

14. Line 126-127. Again Fig. 4A is referred to in the text before Fig. 3.

15. Line 181-182. It is very difficult to see the Q at the Qp site in Figure 3D. Maybe if bolder colors and more contrast was used it would make the Q easier to see.

16. Line 198-200. It is not obvious from Fig. 2B, 2D, how SDHTT5 interacts with NDUB11 and NDUTT5 since the later proteins do not seem to appear in those Figures.

17. Is the thioether linkage at C321 in SDHTT1 how the authors identify the heme as a cytochrome c? See lines 187-188. Also note the absorption spectroscopy in the recent study noted as Ref. 36 does not necessarily confirm that this is a c-type cytochrome.

18. Line 253. This sentence is incomplete. It should say....the physiological significance of CII incorporation into.....(line 252, succinate is the substrate of CII, so "its" in Line 253 is referring back to succinate not CII.)

19. Lines 281-283. How easy is it to distinguish between the densities of cardiolipin and quinone and its isoprenoid tail at the resolution the authors are working with?
20. The bL hemes are not shown in Fig. S8A, (see Lines 282-283).
21. In Fig. S8B the authors should have an arrow pointing to the “protein obstruction” they refer to in the mammalian CIII2
22. Line 300. What is the distance between the two bL hemes?
23. Line 350. Do the authors mean NDUA4’s proposed role inhibiting CIV dimerization.....?
24. Line 395. I believe the authors mean weighed rather than weighted.
25. Line 406-407. Were the 100-KDa Millipore concentrators used for all concentration steps? This is not entirely clear in the paragraph on supercomplex purification.
26. Although the authors have prepared two supplementary videos of the two complexes analyzed which are nice to look at they have never been mentioned in the text, nor is there any comment as to what one should look for in the videos. It seems there should be at least some mention in the text and also some sort of legend for these videos.
27. There is little to no mention of Supplementary Figures 6 and 9 in the text.

Response to reviewers' comments

Reviewer #1 (Remarks to the Author):

The manuscript by Han et al. reports cryo-EM structures of a respiratory “megacomplex”. The manuscript is not easy to follow and it is rather descriptive. Many terms are incorrectly used. The authors do not present any activity measurements of the components of the "megacomplex". Such measurements must be included, otherwise the work shows only a series of structures, but we don't know if these structures represent active enzymes. Is there a "megacomplex" activity?

We thank the reviewer for pointing out the writing and formatting issues of this manuscript, which we have made major efforts to address. Briefly, among other re-writings done, we reformatted the section ‘Organization of Tt-MC IV₂+(I+III₂+II)₂’ (page 6 line 147, please note that line numbers don’t function properly with track changes, therefore the indicated number lines here and below may not be accurate) to state in a clear manner 1) structural similarities and differences between the architectures of mammalian and Tetrahymena megacomplexes; 2) the logical linkage between Tt-CIV₂'s own structural divergence and its contribution to the divergent organization of Tt-MC IV₂+(I+III₂+II)₂ and 3) discussion on the relation between the cyt *c* redox site distance of CIII₂-CIV and cyt *c* turnover kinetics. We also added a section in the Results titled ‘Ciliate-specific subunits of Tt-CIV₂ contribute to megacomplex assembly’ (page 11 line 296) to discuss more in-depth the ciliate-specific subunits and their contributions to the formation and structural integrity of Tt-megacomplex, apart from what was already been included in Zhou et al., 2022, Science. Moreover, we reformatted the section ‘Symmetry re-establishment of Tt-CIII₂ in Tt-MC IV₂+(I+III₂+II)₂’ (page 13 line 338) to state more clearly that how Tt-CIII₂'s symmetry breaking and re-establishment related to its Q-cycle based electron transportation function. We also re-wrote the last two paragraphs (page 13 line 362) to more accurately discuss the plausible QH₂ channelling between Tt-CII and the distal Q_p site of Tt-CIII₂ amongst current kinetic evidences regarding substrate channelling in mammalian respirasome and SC I+III₂. Additionally, we corrected all mis-used terms.

We performed spectroscopic activity assay of the purified sample used for Cryo-EM data collection, using recombinantly expressed Tetrahymena cyt *c* as electron carrier and both NADH and succinate as substrates. Results of these activity assays were included as panels D-H in the current Fig. 1. The assays showed the sample had expected NADH:O₂ and succinate:O₂ oxidoreductase activities, inhibitable by specific inhibitors of CI-CIV including rotenone, malonate, antimycin A and sodium azide. Together with the kinetic curves of Tt-cyt *c* reduction and oxidation monitored at 550 nm, these confirmed that the Tt-megacomplex was capable of electron transfer via both CI-CIII₂-CIV and CII-CIII₂-CIV pathways, a.k.a. megacomplex activities. We also performed spectroscopic activity assays in presence of either NADH, succinate or both NADH and succinate to kinetically investigate the possible QH₂ channelling between Tt-CII and Tt-CIII₂ within the megacomplex.

Specific comments:

L28 What is "oxidative energy"?

We thank the reviewer for pointing out this mis-used term. We aimed to express that the oxidation of NADH and succinate by ETC were energetically coupled to the formation of proton electrochemical gradient

across the inner mitochondrial membrane. The sentence (page 1 line 28) has been changed to ‘The energetically favourable oxidation of NADH or succinate by the electron transport chain (ETC) complexes I-IV (CI-CIV)’.

L30 "proton motive force"

We thank the reviewer for pointing out this typo. It has been corrected accordingly.

L76 What is "conformational rigidity". The term must be defined before use.

We thank the reviewer for pointing out this mis-used term. We aimed to express that the overall conformations of Tt-CII remained mostly constant between Tt-SC IV₂+I+III₂+II and Tt-MC IV₂+(I+III₂+II)₂. The sentence (page 3 line 99) has been changed to ‘A well-resolved 3.26 Å map was produced (Supplementary Figs. 4 and 5G, Supplementary Table 1), indicating that Tt-CII adopts similar conformations in the two assemblies. Focused refinement maps were then combined into a composite map of dimeric Tt-MC IV₂+(I+III₂+II)₂’.

Fig. 1 legend. The authors refer to "diffusion direction". There is no diffusion direction for *cyt c*. It diffuses randomly in the water phase. Do the arrows show a sequence of reactions?

We thank the reviewer for pointing out the incorrect phrasing of ‘diffusion direction’, all relevant phrases in legends and discussions relating to diffusion have been reformatted accordingly. Linking two sequential reaction sites of certain electron carrier by arrow has been used in literatures (Zhou et al., 2022, Science), nonetheless arrows in Fig.1 has been more accurately adjusted to only linking substrates and their corresponding products.

L123 Differences between the current structure and that from mammalian mitochondria are discussed and attributed to "highly divergent Tt-CIV₂". Details of these differences must be discussed, otherwise the statement has no meaning.

Similarities and differences between the architectures of mammalian and Tetrahymena megacomplexes are elaborated in the first paragraph of the section ‘Organization of Tt-MC IV₂+(I+III₂+II)₂’ (page 6 line 147). Briefly, the organization of mammalian SC I+III₂ is conserved in Tetrahymena ETC, but the position of CIV moves from adjacent to the CI MA toe in mammalian respirasome to the side of CI MA heel opposing its Q tunnel in Tetrahymena megacomplex. Moreover, CII is incorporated into Tetrahymena but not mammalian megacomplex.

The relation between Tt-CIV₂'s position in the megacomplex and its own structural divergence compared to mammalian CIV is discussed in the second paragraph (page 6 line 160). Briefly, the sheer size augmentation of Tt-CIV₂, the burial of canonical bridging subunits at Tt-CIV's interface to Tt-CI MA toe and Tt-CIII₂, as well as the Tt-specific ‘toe bridge’ structure embracing the distal protomer of Tt-CIII₂ all necessitate the relocation of Tt-CIV₂ within Tetrahymena megacomplex compared to its mammalian counterpart.

L134- Distances for transfer of *cyt c* are discussed, but it is not clear why. The discussion is descriptive. The authors must elaborate on the consequences of the measured distances on the functionality.

The third paragraph of this section has been re-written to address this issue (page 6 line 174). Decreasing the distance between *cyt c* redox sites on CIII₂ and CIV via supercomplex formation can provide kinetic advantages in electron transfer as demonstrated by theoretical (Stuchebrukhov et al., 2020, Biochimica et

biophysica acta. Bioenergetics), mutational (Berndtsson et al., 2020, EMBO reports) and structural (Moe et al., 2021 PNAS) evidences. The shortest distance between Tt-cyt *c* reduction sites on Tt-CIII₂ and oxidation sites on Tt-CIV₂ is comparable to those in mammalian, yeast and plant SC III₂+IV_{1/2}, confirming similar kinetics in cyt *c* diffusion.

L177-79 Difficult to understand. What is "WT", "electron capacitor"?....

The term 'WT' has been changed to 'wild-type' here page 9 line 249) and also in Methods section 'Cell culture and mitochondrial purification' (page 17 line 474). The term 'electron capacitor' has been changed to 'temporary Q reduction electron sink' (page 10 line 251) as in the cited references (Yankovskaya et al., 2003, Science).

L186-190 The authors speculate on "structural role" vs. participation in electron transfer. The activity of the complex (electron transfer) can be measured. The authors should do the measurement rather than presenting speculations from which the reader cannot draw any conclusions.

The non-canonical heme, coordinated by the second hydrophilic domain of Tt-CII and tentatively assigned as type-c heme, is located beyond direct electron transfer distance to any prosthetic group involved in CII's electron pathway from succinate to Q, therefore cannot participate in its redox function. Tt-CII's activity has been measured within the megacomplex and presented in Fig.1 panels G and H, however to specifically measure electron flux contributed by this non-canonical heme alone seems technically too sophisticated and beyond the scope of this publication. Nonetheless, the sentence (page 10 line 265) has been changed to 'It locates far away from the iron-sulfur clusters beyond direct electron transfer distance (Fig. 4E), therefore not likely involved in electron transportation from succinate to Q by Tt-CII.'

L217-230 The discussion doesn't make sense. Complex III transfers protons across the membrane by a Q-cycle mechanism. The statements here suggest that this is not the case. Please clarify. This section is also difficult to read and understand so I may have misunderstood the conclusions.

This section (page 13 line 339) has been re-written to address issues raised by the reviewers. Briefly, a ~60° rotation of the UQCRFS1 Rieske head containing the Fe₂S₂ cluster, a key step of electron transfer from Q_p site to cyt *c*₁ in the Q-cycle, is sterically blocked for the distal CIII₂ protomer in Tt-SC I+III₂. Therefore only the proximal Tt-CIII₂ protomer is capable of carrying out the Q-cycle based electron and proton transfer. In Tt-MC IV₂+(I+III₂+II)₂ however, both Rieske heads are functional judged by the lack of clear electron densities, thereby functional symmetry between the two Tt-CIII₂ protomers are re-established.

Complexes II and III are presented in the main text, but a discussion on complex IV is missing. Please add.

A section titled 'Ciliate-specific subunits of Tt-CIV₂ contribute to megacomplex assembly' (page 11 line 296) has been added to address this issue. Most structural features of the highly divergent Tt-CIV₂ have already been discussed in Zhou et al., 2022, Science. Here in this section the ciliate specific subunits of Tt-CIV₂ are categorized into three groups and discussed accordingly. They are involved respectively in formation of the highly augmented CIV dimer interface, Tt-CIV₂'s interface to Tt-CI MA heel and Tt-CII, as well as Tt-CIV₂'s charge-swapped cyt *c* crater.

Reviewer #2 (Remarks to the Author):

Han et al. present a structural study of the mitochondrial respiratory megacomplexes of the ciliate model organism *Tetrahymena thermophila*. In their study, the authors reveal the architectures of two further megacomplexes that adapt to the curvature of the cristae membrane. This is a crucial question that has remained elusive for years in what way the respiratory megacomplexes affect mitochondrial morphology and the mechanism that results from the composition and architecture of these molecular machines. In this manuscript, the authors present ~8 MDa megacomplex $IV_2+(I+III_2+II)_2$, as well as a ~10.6 MDa megacomplex $(IV_2+I+III_2+II)$. Both complexes reveal different structural rearrangements. In the first structure CIV₂ associates one copy of supercomplex I+III₂ and one copy of CII to shape the cristae membrane curvature. The other arrangement aims the proximity between CIV₂ and CIII₂ cytochrome *c* binding sites. Taking together, the authors' analysis defines new details of the divergence in eukaryotic electron transport chain organization and how it is related to mitochondria morphology and function.

The manuscript is well written and gives an expanded view of existing studies from super- and megacomplexes regarding their architectures, function and evolution. References are sufficient and SI figures are all necessary to support the current manuscript. I recommend to publish the manuscript in Nature Communication. However, I have only some minor points.

1) Labels need to be better explained in the figure legends. For example, at Fig. 3C. What is SC102 and SB 404? Because it is not listed in the main text either.

We thank the reviewer for pointing out this typo. The cardiolipin and ubiquinone in the current Fig. 4C (original Fig. 3C) were mistakenly labelled by their chain ID + residue number in the model. Their labels have now been updated.

2) The authors should include scale bars for the raw micrographs in sFig 3.

We thank the reviewer for pointing out this inaccuracy. Scale bars have been added for the raw micrographs in the current sFig. 4 and sFig. 6.

3) Would it be possible to show the angular distribution of particles at least for the two megacomplexes in sFig 4?

Angular distributions have been added for the current global and focused refinement resolutions figures sFig. 5 and sFig. 7.

4) The authors should present side chains in sFig 6 M & N.

Side chains have been added for the current sFig. 12 M and N.

5) In the legend of sFig. 9 "Tt-CI" has to be changed into "Tt-CII".

We thank the reviewer for pointing out this typo. This has been corrected in the current sFig. 11.

6) Are the ratios of the mammalian and ciliate supercomplexes in sFig. 10 A & B correct? It is also difficult to differentiate complexes I - IV.

The scale/ratios between mammalian and ciliate SCs in the current sFig. 10 A and B are correct. This can be judged by comparing the conserved portions of CI (blue) in panel A. The colours for the conserved

regions of CI-CIV are changed according to Fig. 1 to better differentiate the individual complexes.

7) The publication from Mühleip et al. (2022) should already be cited in the introduction as common points are addressed in the manuscript.

This publication has now been cited specifically in the last paragraph of the Introduction (page 2 line 61) stated as ‘Our Tt-MC $IV_2+(I+III_2+II)_2$ structure agrees with a recent study of the ~5.8 MDa Tt-SC $IV_2+I+III_2+II$ on multiple aspects including overall organization, membrane curvature adaptation, Tt-CII structure and inter-complex interaction sites (Mühleip et al. 2022).’

8) I found no references to Supplementary Tables 1 and 2 in the text.

Supplementary Table 1 is now cited in the second paragraph of the section ‘Overall structures of T. thermophila ETC megacomplexes’ (line 93, 96, 100, 102) wherever Cryo-EM resolutions and model building of the two megacomplexes are mentioned. Supplementary Table 2 is now cited in wherever subunit identification and nomenclature of ciliate ETC are mentioned in the text (line 106, 157, 166, 177, 199, 202, 209, 260, 269, 271, 274, 276, 299, 302, 330, 355).

Reviewer #3 (Remarks to the Author):

NCOMMS-22-41838-T

This manuscript describes a Cryo-EM description of an approximately 8 and 10.6 MDa supercomplex of the mitochondrial respiratory chain from the ciliate *Tetrahymena thermophila*. The scope of the work is very impressive, however, the manuscript seems to have been hastily put together and it would seem a significant revision is necessary to make it more accessible to the reader. In mammals the mitochondrial respiratory supercomplex is usually composed of complexes I, III, and IV. In humans, however, it has been modeled that a large gap between complexes I and IV may allow complex II to insert into the supercomplex (this is modeled in Ref. 10). The current study clearly supports the notion that in *T. thermophila* that complex II is part of these megacomplexes. Additionally in *T. thermophila* many additional subunits are associated with the respiratory complexes such that 364 individual subunits (compared to about approximately 75 individual subunits in mammalian supercomplexes) have been identified defining the entire electron transport chain of *T. thermophila*. It is curious that the authors do not provide any suggestions for why these many additional subunits are needed for the respiratory chain of *T. thermophila*. A brief discussion would be useful. It is realized that there is likely no definitive answer for why there are so many additional subunits in the single cell ciliate, however reasonable speculation would be appreciated. It should be noted that there is some discussion by the authors regarding the potential role of certain ciliate specific subunits in the Discussion (lines 334-359) regarding the relationship to mammalian supercomplex assembly factors. This sort of information is useful but it seems like it is really part of the Results and not necessarily for the Discussion. Overall the manuscript provides very interesting information about the ciliate respiratory supercomplex but a more detailed analysis of the data and data analysis would make this a much more useful contribution.

We thank the reviewer for pointing out the writing and formatting issues of this manuscript, which we have made major efforts to address. Most of the divergent structural features and structural/functional descriptions regarding individual Tt-CI and Tt-CIV₂ subunits have already been included in Zhou et al., 2022, Science. Nonetheless, we added a section in the Results titled ‘Ciliate-specific subunits of Tt-CIV₂ contribute to megacomplex assembly’ (page 11 line 296) to comprehensively discuss how the additional subunits of Tt-CIV₂ could influence the formation of Tt-MC IV₂+(I+III₂+II)₂. In this section, they were categorized into three groups involved respectively in formation of the highly augmented Tt-CIV dimer interface, Tt-CIV₂’s interface to Tt-CI MA heel and Tt-CII, as well as Tt-CIV₂’s charge-swapped cyt *c* crater. We also added a section in the supplementary discussion titled ‘Ciliate-specific subunits of Tt-CI’ (page 1 line 23) to discuss how the additional subunits of Tt-CI contributed to Tt-MC IV₂+(I+III₂+II)₂ assembly, apart from what had already been included in Zhou et al., 2022, Science. Regarding the supercomplex assembly factors, we moved the description of NDUA4, which was actually involved in Tt-CIV dimerization, to the newly added section in Results. Meanwhile homologues of SCAF1 (COX7A) and Rcf2 (COXTT17) were not likely genuine assembly factors in Tetrahymena ETC and their contents were kept in Discussion.

Specific comments: These are listed in order as they appear in the text but some are much more significant issues than others.

1. Page 2, line 54. In Supplementary Fig. 1, line 27; do the authors mean the pooled fractions from B and F or do they mean from A and E. The green boxes indicating the samples used for Cryo-EM appear in B and F, but also the blue hatching in A and E seem to indicate specific fractions although that just may refer to the gel(s) below.

Here we refer to the pooled fractions indicated by the green boxes in gels of panel B and F. The blue hatching is to indicate fractions collected in gradient ultracentrifugation and chromatography and loaded onto the gels in panels B, D, F and H. To avoid confusion, the hatching has been changed to yellow.

2. Also regarding Fig. S1. Although the authors clearly show in their structure that complex II is part of the “megacomplex” none of the bands in Fig. S1, B, D, F, or H ever indicate the presence of complex II. It must be somewhere in the gel? One assumes that the samples labeled Ti-MC or Ti-Sc I, III, IV must have also included complex II? Is that the case? Please clarify.

We thank the reviewer for pointing out the typos regarding labels of gel bands to the left of panels B, D, F and H of Fig. S1. Now the labels have been corrected to ‘Tt-MC IV₂+(I+III₂+II)₂’ and ‘Tt-SC IV₂+I+III₂+II’.

3. Supplementary Fig. 2 legend. Please define what the pinkish region is supposed to represent since the same color seems to be used to represent different complexes and different regions of the same complex.

The transparent pink regions in the current Fig. S4-S7 indicate masks used for each overall and focused refinements during Cryo-EM image processing. This has been made clear in legends of these figures.

4. Line 73. The authors begin discussing Supplementary Fig. S4 before they discuss Supplementary Figure S3. Why not swap Figs. S3 and S4? They should re-order and renumber their figures so that the reader doesn’t have to hunt around for the figure. It is not always clear which figure the authors are referring to or if they have made a typo in the text.

We thank the reviewer for pointing out the ordering issues of the supplementary figures. We’ve now re-ordered these figures according to their order of appearance in the text. We’ve also added two supplementary

figures of Tt-cyt *c* expression (currently Fig. S2) and megacomplex activity measurements (currently Fig. S3), which were also placed in order along with others.

5. Line 83 and 84. The authors bring up the idea that membrane arm (heel) of complex I is opposite the side of the ubiquinone (Q) tunnel. Since they have never discussed the Q tunnel to this point of the manuscript this is hard to follow. Also no Q-tunnel is shown in Fig. 1 so maybe this comment is premature or the phrasing needs to be altered.

We added a sentence (page 6 line 149) to describe the term ‘Q tunnel’ at the beginning of the section ‘Organization of Tt-MC IV₂+(I+III₂+II)₂’ stating ‘The tunnel for Q/QH₂ to enter and exit CI opens to the concave side of CI MA bordered by CIII₂ in SC I+III₂ (Fig. 1C)’. Moreover, we now show the position and shape of mammalian CI Q tunnel by red surfaces in Fig. 1 C and update the legends accordingly.

6. The colors chosen to represent CII and CIV in Fig. 1A are quite similar and makes it hard to distinguish the subunits. The bolder colors in Fig. 1B (dark red for CII, magenta for CIV) for these two complexes might make a much sharper figure where it is easier to discern the boundaries between these complexes. The same comment can be made about Fig. 2.

We used transparent atomic surfaces in Fig. 1A and Fig. 2 so that atomic details of the subunits/molecules of interests could be seen through the backgrounds of each complex. The individual complexes were color-coded as in the previous publication Zhou et al, 2022, Science. We’ve changed the color of Tt-CII to orange and adjusted the overall levels of transparency to better distinguish between different complexes.

7. Line 85. It is actually Ref. 10 (not 24 as indicated in the text) where the “wedge” between CI and CIV is suggested as a site for CII to bind.

This sentence together with other detailed descriptions regarding the organizations of individual complexes within the megacomplex has been moved to the next section ‘Organization of Tt-MC IV₂+(I+III₂+II)₂’ (page 6 line 147). The wrong reference has been corrected to Guo et al., 2017 Cell (current ref 11).

8. Line 97. Once again a supplementary figure is discussed out of order. Fig. S7 before any mention of Fig. S6. Fix this.

We’ve now re-ordered these figures according to their order of appearance in the text.

9. Line 104. Fig. S10 C and D discussed before any mention of Figs. S6, 8 or 9.

We’ve now re-ordered these figures according to their order of appearance in the text.

10. Line 100. Please confirm that Reference 15 is the proper reference for the term “respiratory patch”.

We’ve changed this sentence (page 4 line 123) to ‘This provides a glimpse into the structure of ciliate respiratory patch and suggests that it contains at least stacks of ETC half rings that sheath the tubular cristae.’ and cited the proper reference describing the ciliate respiratory patch (Allen et al., 1989 The Journal of cell biology, current ref 16). The Nübel et al., 2009 Proteomics (previous ref 15, current ref 18) described possible organizations of ‘respiratory patch’ in the yeast *Yarrowia lipolytica*. We’ve also added a few more references including Bultema et al., 2009, Biochimica et biophysica acta (ref 13), Wittig et al., 2006, Biochimica et biophysica acta (ref 17) and Strecker et al., 2010, Proteomics (ref 19) describing models of ‘respiratory patch’

or 'respiratory string' in a range of species. We've now moved the first mentioning of 'respiratory patch' to the first paragraph of Introduction (page 2 line 40) with these references cited.

11. Line 118. The authors need to refer to their Supplementary Table 2 when they begin to discuss the individual subunits of the complexes and especially when referring to the many additional subunits from the ciliate.

Supplementary Table 2 is now cited here and wherever subunit identification and nomenclature of ciliate ETC are mentioned in the text (line 106, 157, 166, 177, 199, 202, 209, 260, 269, 271, 274, 276, 299, 302, 330, 355).

12. Table S2. It would be useful to indicate a reference for the assignment of the role of the proteins listed in Table S2. For example, in Line 118 it is stated that COXMC3 is a mitochondrial carrier subunit. How/who determined this? If it is just how it is annotated in databases please indicate which database was used.

The reference Zhou et al., 2022, Science has been added for Table S2 and at the beginning of the second paragraph of Introduction (page 2 line 50). This reference describes in detail the assignments, structures and possible functional roles of individual subunits of Tt-CI, Tt-CIII₂ and Tt-CIV₂. It also describes how the three mitochondrial carrier subunits of Tt-CIV₂ (COXMC1-3) are assigned based on annotations in UniProt and conservation of key substrate contacting residues in the central cavity. A brief discussion has been added in the second paragraph of the section 'Ciliate-specific subunits of Tt-CIV₂ contribute to megacomplex assembly' (page 12 line 311) on the annotation of COXMC1 and 2.

13. Line 119 and 120. It is not obvious from Fig. 2A that COXMC1 interacts with CII. Again the color contrast in the figure makes this point very obscure.

The background colour for Tt-CII has been changed to orange to improve contrast in Fig. 1 and 2. In the current Fig. 2A COXMC1 (cyan) borders the interface between Tt-CIV₂ and Tt-CII.

14. Line 126-127. Again Fig. 4A is referred to in the text before Fig. 3.

The order of Fig. 3 and 4 has been switched.

15. Line 181-182. It is very difficult to see the Q at the Qp site in Figure 3D. Maybe if bolder colors and more contrast was used it would make the Q easier to see.

In the current Fig. 4D Q has been updated to bright yellow to improve the contrast.

16. Line 198-200. It is not obvious from Fig. 2B, 2D, how SDHTT5 interacts with NDUB11 and NDUTT5 since the later proteins do not seem to appear in those Figures.

In the current Fig. 2D SDHTT5 (orange), NDUB11 (magenta) and NDUTT5 (cyan) are all present to show their interactions.

17. Is the thioether linkage at C321 in SDHTT1 how the authors identify the heme as a cytochrome c? See lines 187-188. Also note the absorption spectroscopy in the recent study noted as Ref. 36 does not necessarily confirm that this is a c-type cytochrome.

The thioether linkage is the major evidence based on which we identify this non-canonical heme as type-c, since such linkage is absent in type-a or b hemes. The authors now understand that the spectrum deconvolution in the current ref 28 (previous ref 26) is not compelling evidence of the presence of type-c

heme, therefore the sentence has been re-phrased as ‘we tentatively assign this non-canonical heme as a heme c.’ (page 10 line 264).

18. Line 253. This sentence is incomplete. It should say....the physiological significance of CII incorporation into.....(line 252, succinate is the substrate of CII, so “its” in Line 253 is referring back to succinate not CII.)

This sentence has been corrected accordingly (page 14 line 384).

19. Lines 281-283. How easy is it to distinguish between the densities of cardiolipin and quinone and its isoprenoid tail at the resolution the authors are working with?

The content regarding the wedged Q has been moved to supplementary discussion (page 2 line 46) to keep the logical flow of the main text concise and clear. At current resolution of Tt-CII (3.26 Å) cardiolipin and quinone can be easily differentiated since the former molecule had its characteristic density of diphosphatidylglycerol with four alkyl tails. The sentence (page 2 line 50) in the supplementary discussion has been re-phrased accordingly.

20. The b_L hemes are not shown in Fig. S8A, (see Lines 282-283).

The b_L hemes are now shown in the current Fig. S14A (previous Fig. S8A).

21. In Fig. S8B the authors should have an arrow pointing to the “protein obstruction” they refer to in the mammalian CIII₂

The protein obstruction between the two Q cavities of mammalian Tt-CIII₂ is now labeled with arrow in the current Fig. S14A (previous Fig. S8A).

22. Line 300. What is the distance between the two b_L hemes?

The sentence has been moved to supplementary discussion (page 3 line 70) and re-phrased to ‘Additionally, its position in middle of the two b_L hemes implicates a role as a single-electron relay station compensating the 14.5 Å distance beyond direct electron transfer between them.’

23. Line 350. Do the authors mean NDUA4’s proposed role inhibiting CIV dimerization.....?

Yes. The sentence (page 16 line 438) has been re-phrased to ‘In the ciliate respiratory chain however, its proposed role inhibiting CIV dimerization can be ruled out due to its presence on the dimer interface of Tt-CIV₂.’

24. Line 395. I believe the authors mean weighed rather than weighted.

Yes. It has been corrected (page 17 line 487).

25. Line 406-407. Were the 100-KDa Millipore concentrators used for all concentration steps? This is not entirely clear in the paragraph on supercomplex purification.

Yes. The sentence (page 17 line 497) has been changed to ‘The supernatant was concentrated to ~3 ml in a centrifugal concentrator with 100 KDa molecular weight cut off (MWCO) (Millipore), which was used for all following concentration steps in this section.’

For the newly added Methods section ‘Tt-cytochrome c (cyt c) expression’, 3KDa MWCO centrifugal concentrators were used to concentrate recombinantly expressed Tt-cyt c (page 18 line 520).

26. Although the authors have prepared two supplementary videos of the two complexes analyzed which are nice to look at they have never been mentioned in the text, nor is there any comment as to what one should look for in the videos. It seems there should be at least some mention in the text and also some sort of legend for these videos.

Supplementary Video 1 and 2 have now been mentioned in page 4 line 106 and page 4 123 of the main text. A legend has been added for these two videos in page 32 line 223 of the supplementary information.

27. There is little to no mention of Supplementary Figures 6 and 9 in the text.

Fig. S6 and S9 have been updated to Fig. S12 and S11, respectively. Fig. S12 has now been mentioned in the main text line 247, 250, 253, 349, 351 and 375. Fig S11 has now been mentioned in the main text line 237.

REVIEWERS' COMMENTS

Reviewer #2 (Remarks to the Author):

I have once again dealt intensively with the manuscript, supplement and rebuttal. All of my concerns were addressed and resolved. However, I have some minor comments on revised passages in the manuscript. When these improvements have been made, I recommend the manuscript for publication by Nature Communications.

1) Line 241-244. Regarding the activity assays, I don't understand the differences in Fig. 1 E&F and G&H, because they basically show the same measurements with NaN_3 . These should at least be made clear in the figure legend.

2) Line 437-440. If one considers the nomenclature, the authors should also write B- and C-type in small letters.

3) Line 601. In my opinion, the distance between the bL hemes should be sufficient for direct electron transfer. The edge-to-edge distance is still in the 14 Å limit, which refers to electron tunneling rates between metal centers in proteins and allow efficient electron transfer according to the generally accepted 'Moser-Dutton ruler' (Page et al., 1999). I may have overlooked this for complex III. However, please clarify this statement.

Reviewer #3 (Remarks to the Author):

NCOMMS-22-41838A

The revised submission by Han et al., is much improved but there are still a few minor areas for improvement that if addressed would improve the manuscript.

1. On page 2, line 64 the authors seem to state that no structure of eukaryotic complex II is available. However, this is not correct as there are a number of structures of the porcine complex data in the pdb

database. The initial CII porcine structure was by Sun et al., Cell 121: 1043-1057 (2005) although there are other more recent eukaryotic structures also available. What is shown in the current manuscript is the first structure of the Tetrahymena CII, and in a supercomplex, maybe that is what the authors mean.

2. On page 3, line 124 the authors should list in the text the activity they measure for the succinate dependent activity. In the line above the list the NADH dependent activity as about 400 nmole/min/mg protein but it is difficult to tell from Fig. S3 what they are suggesting is the succinate dependent activity. It would seem from Fig. 1G that the activity is around 45 nmol/min/mg protein when one subtracts the malonate insensitive activity. Is that correct?

3. A minor note is that several times in the text the authors stateelectron transportation activity..... I think they mean....electron transport activity.....? This is the more common phrasing.

4. Page 13, line 602. The authors seem to state that an electron transfer distance of 11 angstroms is too far for electron transfer to occur. This is not correct. Based on numerous studies (see those of P. L. Dutton et al) electron transfer can occur proficiently at up to 14 angstroms distance so it is only beyond that distance that it would be unlikely that electron is transferred between the two CIII protomers. If the distance is 14.5 angstroms the distance may suggest that there is no electron transfer between the two protomers of CIII, however, as there is likely some error in the distances measured in the supercomplex the authors should be careful of making hardfast statements.

5. In the first paragraph on page 3 of the revised manuscript the authors appear to mention that the lower oxidation rate of NADH shown in Figure 1D is because too much enzyme (10 nM) was used for the assay resulting in not capturing the initial rate. This is supported by the data seen in Supplementary 3C. It is further commented upon that using 2 nM of enzyme allows capture of the initial rate and you then see a higher NADH oxidation rate of about 400 nmol/min/mg protein which is more in line with what is expected. The authors, however, never mention in the "Methods" section (page 20) is describing the assay that they have made this correction and they state they used 10 nM of the Tt-MC complex in each assay. This should be explained more clearly so as not to confuse the reader. It is never stated in the Figure legends (or Fig. S3 legend) that different amounts of protein are used in the assays. Otherwise the assays appear competently done and appropriate inhibitors have been used. Please state explicitly in the Figure S3 legend that different amounts of protein 2 nM (Fig. S3A and apparently S3B), have been used and in Fig. S3C it is 10 nM of Tt-MC.

Response to reviewers' comments

Reviewer #2 (Remarks to the Author):

I have once again dealt intensively with the manuscript, supplement and rebuttal. All of my concerns were addressed and resolved. However, I have some minor comments on revised passages in the manuscript. When these improvements have been made, I recommend the manuscript for publication by Nature Communications.

1) Line 241-244. Regarding the activity assays, I don't understand the differences in Fig. 1 E&F and G&H, because they basically show the same measurements with NaN₃. These should at least be made clear in the figure legend.

We thank the reviewer for pointing out this inaccuracy and have modified the figure legend to make this clear. For NADH:O₂ and succinate:O₂ oxidoreductase activities measured at 550 nm, cyt *c* reduction rates are calculated as the slopes of the initial linear increasing phases between 50s and 150s in Supplementary Fig. 3C right panel and 3D, respectively. Cyt *c* oxidation rates are calculated as the slopes of the last linear decline phases in Supplementary Fig. 3C right panel and 3D, respectively. Since full reduction of cyt *c* take different amounts of time under different treatments, time windows for this last linear decline phases are defined differently, as described in detail in the Methods. Briefly, for NADH:O₂ oxidoreductase activities in Supplementary Fig. 3C right panel, 1200s-1500s is used for the control and sodium azide curves while 3000s-3300s is used for the blank curve. For succinate:O₂ oxidoreductase activities in Supplementary Fig. 3D 5200s-5500s is used for the control, sodium azide and blank curves.

2) Line 437-440. If one considers the nomenclature, the authors should also write B- and C-type in small letters.

We thank the reviewer for pointing out this inconsistency. We have changed them to lower cases.

3) Line 601. In my opinion, the distance between the bL hemes should be sufficient for direct electron transfer. The edge-to-edge distance is still in the 14 Å limit, which refers to electron tunneling rates between metal centers in proteins and allow efficient electron transfer according to the generally accepted 'Moser-Dutton ruler' (Page et al., 1999). I may have overlooked this for complex III. However, please clarify this statement.

We acknowledge the reviewer's point about the 14 Å limit of direct electron tunneling and have corrected the main text accordingly. The edge-to-edge distance between the two bL hemes in our Tt-MC IV₂+(I+III₂+II)₂ structure, measured between the C2B carbon atoms of the two porphyrin rings as in the current ref 59, is 17.5 Å. The same edge-to-edge distance is 13.3 Å in the recent 2 Å resolution structure of *Arabidopsis thaliana* SC I+III₂ (PDB 8BPX). Therefore, direct electron tunneling cannot happen between *T. thermophila* CIII₂ protomers, but can happen between *A. thaliana* CIII₂ protomers in respective supercomplex environments. PDB atomic nomenclature of ligand HEM is used here for heme b. Main text here has been updated according to above descriptions.

Reviewer #3 (Remarks to the Author):

NCOMMS-22-41838A

The revised submission by Han et al., is much improved but there are still a few minor areas for improvement that if addressed would improve the manuscript.

1. On page 2, line 64 the authors seem to state that no structure of eukaryotic complex II is available. However, this is not correct as there are a number of structures of the porcine complex data in the pdb database. The initial CII porcine structure was by Sun et al., Cell 121: 1043-1057 (2005) although there are other more recent eukaryotic structures also available. What is shown in the current manuscript is the first structure of the Tetrahymena CII, and in a supercomplex, maybe that is what the authors mean.

We thank the reviewer for pointing out this inconsistency. We are aware of the existing eukaryotic CII structures, as we have cited the 2005 Cell paper about the porcine CII structure and the 2006 JBC paper about the avian CII structure later in the Tt-CII section. However, we recognize that our expression here is misleading as if there is currently no eukaryotic CII structure. We have re-phrase this sentence to make it clear that existing CII structures are limited to metazoans, while CII structural diversity in other eukaryotic clades is yet to be studied.

2. On page 3, line 124 the authors should list in the text the activity they measure for the succinate dependent activity. In the line above the list the NADH dependent activity as about 400 nmole/min/mg protein but it is difficult to tell from Fig. S3 what they are suggesting is the succinate dependent activity. It would seem from Fig. 1G that the activity is around 45 nmol/min/mg protein when one subtracts the malonate insensitive activity. Is that correct?

We agree with the reviewers that the malonate-sensitive succinate dependent activity is around 47 nmol/min/mg protein, roughly in agreement with the previously reported ~30 nmole/min/mg protein succinate-dependent oxygen consumption rate measured with *T. thermophila* cells. Please note that there is a 1:2 ratio between succinate reduction and oxygen consumption rates.

3. A minor note is that several times in the text the authors state "...electron transportation activity....." I think they mean "...electron transport activity...."? This is the more common phrasing.

We thank the reviewer for pointing out this typo which we have corrected.

4. Page 13, line 602. The authors seem to state that an electron transfer distance of 11 angstroms is to far for electron transfer to occur. This is not correct. Based on numerous studies (see those of P. L. Dutton et al) electron transfer can occur proficiently at up to 14 angstroms distance so it is only beyond that distance that it would be unlikely that electron is transferred between the two CIII protomers. If the distance is 14.5 angstroms the distance may suggest that there is no electron transfer between the two protomers of CIII, however, as there is likely some error in the distances measured in the supercomplex the authors should be careful of making hardfast statements.

We acknowledge the reviewer's point about the 14 Å limit of direct electron tunneling and have corrected the main text accordingly. The edge-to-edge distance between the two b_L hemes in our Tt-MC IV₂+(I+III₂+II)₂ structure, measured between the C2B carbon atoms of the two porphyrin rings as in the current ref 59, is 17.5 Å. The same edge-to-edge distance is 13.3 Å in the recent 2 Å resolution structure of *Arabidopsis thaliana* SC I+III₂ (PDB 8BPX). Therefore, direct electron tunneling cannot happen between *T. thermophila* CIII₂ protomers, but can happen between *A. thaliana* CIII₂ protomers in respective supercomplex environments. PDB atomic nomenclature of ligand HEM is used here for heme b. Main text here has been updated according to above descriptions.

5. In the first paragraph on page 3 of the revised manuscript the authors appear to mention that the lower oxidation rate of NADH shown in Figure 1D is because too much enzyme (10 nM) was used for the assay resulting in not capturing the initial rate. This is supported by the data seen in Supplementary 3C. It is further commented upon that using 2 nM of enzyme allows capture of the initial rate and you then see a higher NADH oxidation rate of about 400 nmol/min/mg protein which is more in line with what is expected. The authors, however, never mention in the “Methods” section (page 20) is describing the assay that they have made this correction and they state they used 10 nM of the Tt-MC complex in each assay. This should be explained more clearly so as not to confuse the reader. It is never stated in the Figure legends (or Fig. S3 legend) that different amounts of protein are used in the assays. Otherwise the assays appear competently done and appropriate inhibitors have been used. Please state explicitly in the Figure S3 legend that different amounts of protein 2 nM (Fig. S3A and apparently S3B), have been used and in Fig. S3C it is 10 nM of Tt-MC.

We thank the reviewer for pointing out this inaccuracy. In the ‘Spectroscopic assays for megacomplex IV₂+(I+III₂+II)₂ activity’ section of the Methods part we have stated, in separate paragraphs, that 10 nM and 2 nM megacomplex samples are used for activity assays in Supplementary Fig. 3cd and Supplementary Fig. 3ab, respectively. We’ve now further emphasized this point by adding corresponding figure citations to respective paragraphs in this section. We’ve also added descriptions in the legends of Fig. 1 and Supplementary Fig. 3 to make them clear.